# Insights into the conservation and diversification of the molecular functions of YTHDF proteins

Daniel Flores-Téllez[1,2], Mathias Due Tankmar[1], Sören von Bülow[1], Junyu Chen[1], Kresten Lindorff-Larsen[1], Peter Brodersen[1]*, Laura Arribas-Hernández[1]*

**1** University of Copenhagen, Biology Department. Copenhagen, Denmark, **2** Universidad Francisco de Vitoria, Facultad de Ciencias Experimentales. Pozuelo de Alarcón (Madrid), Spain

* pbrodersen@bio.ku.dk (PB); laura.arribas@bio.ku.dk (LA-H)

## Abstract

YT521-B homology (YTH) domain proteins act as readers of *N6*-methyladenosine (m$^6$A) in mRNA. Members of the YTHDF clade determine properties of m$^6$A-containing mRNAs in the cytoplasm. Vertebrates encode three YTHDF proteins whose possible functional specialization is debated. In land plants, the YTHDF clade has expanded from one member in basal lineages to eleven so-called EVOLUTIONARILY CONSERVED C-TERMINAL REGION1-11 (ECT1-11) proteins in *Arabidopsis thaliana*, named after the conserved YTH domain placed behind a long N-terminal intrinsically disordered region (IDR). *ECT2, ECT3* and *ECT4* show genetic redundancy in stimulation of primed stem cell division, but the origin and implications of YTHDF expansion in higher plants are unknown, as it is unclear whether it involves acquisition of fundamentally different molecular properties, in particular of their divergent IDRs. Here, we use functional complementation of *ect2/ect3/ect4* mutants to test whether different YTHDF proteins can perform the same function when similarly expressed in leaf primordia. We show that stimulation of primordial cell division relies on an ancestral molecular function of the m$^6$A-YTHDF axis in land plants that is present in bryophytes and is conserved over YTHDF diversification, as it appears in all major clades of YTHDF proteins in flowering plants. Importantly, although our results indicate that the YTH domains of all arabidopsis ECT proteins have m$^6$A-binding capacity, lineage-specific neo-functionalization of ECT1, ECT9 and ECT11 happened after late duplication events, and involves altered properties of both the YTH domains, and, especially, of the IDRs. We also identify two biophysical properties recurrent in IDRs of YTHDF proteins able to complement *ect2 ect3 ect4* mutants, a clear phase separation propensity and a charge distribution that creates electric dipoles. Human and fly YTHDFs do not have IDRs with this combination of properties and cannot replace ECT2/3/4 function in arabidopsis, perhaps suggesting different molecular activities of YTHDF proteins between major taxa.

**Data Availability Statement:** All relevant data is provided within the manuscript, the supplementary information, and the Zenodo repository: https://doi.org/10.5281/zenodo.8269719.

**Funding:** This work was supported by the Novo Nordisk Foundation through a Hallas-Møller Ascending Investigator 2019 grant (NNF19OC0054973) to P.B. and a grant to K.L.-L. via the PRISM (Protein Interactions and Stability in Medicine and Genomics) center (NNF18OC0033950), by the European Molecular Biology Organisation through a long-term fellowship (ALTF 810-2022) to S.v.B., by the European Research Council through a Consolidator grant (ERC-2016-CoG 726417 PATHORISC) to P. B., and by the Independent Research Fund Denmark through a Research Project2 grant (9040-00409B) to P.B. The funders had no role in study design, data collection and analysis, decision to publish, or preparation of the manuscript. The following authors received salary from the funders: M.D.T. (Novo Nordisk Foundation, grant NNF19OC0054973); S.v.B. (Novo Nordisk Foundation, grant NNF18OC0033950, and European Molecular Biology Organisation, grant ALTF 810-2022); J.C. (Novo Nordisk Foundation, grant NNF19OC0054973); L.A.-H. (European Research Council grant ERC-2016-CoG 726417, Independent Research Fund Denmark, grant 9040-00409B, and Novo Nordisk Foundation, grant NNF19OC0054973).

**Competing interests:** The authors declare that they have no competing interests.

## Author summary

Regulation of gene expression is essential to life. It ensures correct balancing of cellular activities and the controlled proliferation and differentiation necessary for the development of multicellular organisms. Methylation of adenosines in mRNA (m$^6$A) contributes to genetic control, and absence of m$^6$A impairs embryo development in plants and vertebrates. m$^6$A-dependent regulation can be exerted by a group of cytoplasmic proteins called YTHDFs. Higher plants have many more YTHDFs than animals, but it is unknown whether these many YTHDF proteins carry out fundamentally different or roughly the same molecular functions, as only a small fraction of them have been studied thus far. This work addresses the origin and reasons behind YTHDF expansion during land plant evolution and reveals that most YTHDFs in flowering plants have the same molecular functions that facilitate rapid division of differentiating stem cells. Remarkably, this molecular activity is present in the most basal lineage of land plants and functions similarly across 450 million years of land plant evolution. We also identified a few plant YTHDF proteins with divergent molecular function despite their ability to bind to m$^6$A. Our work provides a firm basis for further advances on understanding molecular properties and biological contexts underlying YTHDF diversification.

## Introduction

*N6*-methyladenosine (m$^6$A) is the most abundant modified nucleotide occurring internally in eukaryotic mRNA. It is of major importance in gene regulation as illustrated by the embryonic lethality of mutants in the dedicated mRNA adenosine methyltransferase in higher plants [1] and in mammals [2]. The presence of m$^6$A in an mRNA may have multiple biochemical consequences. These include changes in the secondary structure [3–5] and the creation of binding sites for RNA-binding proteins specialized for m$^6$A recognition [6,7]. YT521-B homology (YTH) domain proteins [8] constitute the best studied class of m$^6$A-binding proteins [6,7,9]. They achieve specificity for m$^6$A via an aromatic pocket accommodating the *N6*-adenosine methyl group, such that the affinity of isolated YTH domains for m$^6$A-containing RNA is 10-20-fold higher than for unmodified RNA [9–14].

Two different phylogenetic clades of YTH domains have been defined, YTHDC and YTHDF, sometimes referred to simply as DC and DF [6,15]. Genetic studies show that major functions of m$^6$A in development in both vertebrates and higher plants depend on the YTHDF clade of readers [16–22]. In all cases studied in detail thus far, plant and animal YTHDF proteins are cytoplasmic in unchallenged conditions [15,16,23–26], and contain a long N-terminal intrinsically disordered region (IDR) in addition to the C-terminal YTH domain [6,8,27]. While the YTH domain is necessary for specific binding to m$^6$A in mRNA [9–14], the IDR is considered to be the effector part of the protein [9, 28–30]. Nonetheless, it has been proposed that the IDR may also participate in RNA binding, because the YTH domain alone has low affinity for mRNA [6]. Indeed, the IDR-dependent crosslinks between a YTHDF protein and mRNA detected upon UV-irradiation of living arabidopsis seedlings [31] experimentally supports such a mechanism, conceptually equivalent to the contribution of IDRs in transcription factors to specific DNA binding [32]. Additionally, the IDR may be involved in phase separation. Plant YTHDF proteins can form condensates *in vitro* [16] and, upon stress, localize *in vivo* to distinct foci [16,26] identified as stress granules [26]. Similar properties have been reported for human YTHDFs, which engage in liquid-liquid phase separation when concentrated on polyvalent m$^6$A-modified target RNA *in vitro* [33–36].

While yeast, flies and primitive land plants encode only one YTHDF protein [24–26,37,38], vertebrates have three closely related paralogs (YTHDF1-3) [6,8,15,39] and higher plants encode an expanded family, with eleven members in *Arabidopsis thaliana* (*Ath*) referred to as EVOLUTIONARILY CONSERVED C-TERMINAL REGION1-11 (ECT1-11) [27,40]. It is a question of fundamental importance for the understanding of how complex eukaryotic systems use the regulatory potential of m⁶A whether these many YTHDF proteins perform the same biochemical function, or whether their molecular properties are specialized. Such molecular specialization could, for instance, arise as a consequence of differential binding specificity to mRNA targets, distinct protein-protein interaction properties, or distinct biophysical properties of IDRs, such as those related to the propensity to phase separate [41,42]. Alternatively, diversification of biological functions could be achieved with distinct expression patterns or induction by environmental cues, even if targets and molecular functions are the same. The studies on YTHDF specialization in vertebrate cells illustrate the importance of the topic, and, equally, the importance of addressing it rigorously [43]. Initial work on mammalian cell cultures advocated a model in which YTHDF1 would enhance translation of target mRNAs, YTHDF2 would promote mRNA decay, and YTHDF3 would be able to trigger either of the two [9,44–46]. Nonetheless, subsequent research work in mouse, zebrafish and human cell culture involving single and combined *ythdf* knockouts and analysis of interacting mRNAs and proteins [18,22,39], and comparative studies of human YTHDF1/2/3 structure and molecular dynamics [47] do not support functional specialization, and propose a unified molecular function for all three vertebrate YTHDFs in accelerating mRNA decay. Whether metazoan YTHDF1/2/3 are molecularly redundant or not, and what their precise molecular functions may be, are topics that continue to be debated as of today [48].

The great expansion of YTHDF proteins in higher plants is unique among eukaryotes [8]. Phylogenetic analyses of plant YTHDF domains have established the existence of 3 clades in angiosperms, DF-A (comprising *Ath* ECT1, ECT2, ECT3, ECT4), DF-B (comprising *Ath* ECT5, ECT10, ECT9), and DF-C (comprising *Ath* ECT6, ECT7, ECT8 and ECT11) [26]. The fact that the eleven arabidopsis paralogs have the aromatic residues necessary for specific binding to m⁶A [49] suggests that they may all function as m⁶A readers. The unified model for YTHDF function recently proposed for the three vertebrate paralogs [18,22,39] is consistent with what had already been established for *Ath* ECT2, ECT3 and ECT4 in the plant DF-A clade [16], even though the three plant paralogs are more divergent in sequence than the highly similar mammalian YTHDF1-3 [8,16,39]. The two most highly expressed members in *A. thaliana*, ECT2 and ECT3, accumulate in dividing cells of organ primordia and exhibit genetic redundancy in the stimulation of stem cell proliferation during organogenesis [16,17]. The two proteins probably act truly redundantly *in vivo* to control this process, because they associate with highly overlapping target sets in wild type plants, and each exhibits increased target mRNA occupancy in the absence of the other protein [23]. Simultaneous knockout of *ECT2* and *ECT3* causes a 2-day delay in the emergence of the first true leaves, aberrant leaf morphology, slow root growth and defective root growth directionality among other defects [16,17] that resemble those of plants with diminished m⁶A deposition [50–52]. For the third DF-A clade member, *ECT4*, the genetic redundancy is only noticeable in some tissues as an exacerbation of *ect2/ect3* phenotypes upon additional mutation of *ECT4*, most conspicuously seen in leaf morphogenesis [16,17]. Despite the strong evidence for redundant functions among these three YTHDF paralogs in the DF-A clade, the presence of many other YTHDF proteins in arabidopsis leaves open the question of whether substantial functional specialization of YTHDF proteins exists in plants.

In this study, we systematically define overlaps in the molecular functions of YTHDF proteins in land plants (embryophytes). Employing functional complementation of leaf formation

in triple *ect2/ect3/ect4* (*te234*) mutants, we demonstrate that at least one member of all clades in arabidopsis, and the only YTHDF protein from a group of primitive land plants, can replace ECT2/3/4 functions. In contrast, a few late-diverging ECTs were not able to perform the molecular functions of ECT2/3/4. Based on these results, we propose an ancestral molecular role of land plant YTHDF proteins in stimulation of primordial cell proliferation, and sustained functional redundancy during the diversification process that started more than 400 million years ago (Mya). In addition, our results also support the functional divergence of a small subset of fast-evolving plant YTHDF proteins with contributions to specialization mainly from the IDRs, but also from the YTH domains.

## Results

### The phylogeny of YTHDF proteins comprises several clades in land plants

The adaptation of plants to terrestrial life was accompanied by the acquisition of morphological, physiological, and genomic complexity [53–55]. Knowing how diversification of YTHDF proteins came about during the course of plant evolution is relevant to understand their distinct functions and may hint at roles they might have played in this process. However, the species included in the so far most detailed phylogenetic study on plant YTHDF proteins jump from bryophytes (mosses, liverworts and hornworts [54,56–60]) to angiosperms (flowering plants) [26]. To include the YTHDF repertoires of intermediately diverging clades of embryophytes such as lycophytes, ferns, and gymnosperms, we performed phylogenetic analyses using YTHDF proteins from 32 species widely spread across land plant evolution. We also included YTHDFs from 8 species of green algae (chlorophytes and charophytes) and, as outgroup, 3 microalgal species from eukaryotic taxa evolutionarily close to plants (Fig 1A). In the resulting phylogenetic tree, all embryophyte YTHDFs branch from a single stem that connects this monophyletic group to the more divergent green algal orthologs (Figs 1B and S1–S3), whose peculiarities are further considered in the discussion. Regarding land plants, the tree largely agrees with the DF-A/B/C clades defined on the basis of angiosperm YTHDF sequences [26], with some minor differences. We consider that the former DF-C group comprising *Ath* ECT6/7/8/11 [26] can be subdivided into two groups that diverged early during YTHDF radiation (Figs 1B and S1). Thus, we introduced the name 'DF-D' for the clade defined by one *Amborella trichopoda* (a basal angiosperm) YTHDF protein (*Atr* DFD) and *Ath* ECT6/7 (Figs 1B and S1). Furthermore, since the group comprising bryophyte YTHDFs did not receive a designation in previous studies, we named it DF-E (mnemotechnic for 'Early'). It includes YTHDF proteins from bryophytes and some lycophyte orthologs that branched close to the moss subgroup (Figs 1B and S1). Finally, an additional group composed of YTHDF proteins from six species of ferns and two gymnosperms was named DF-F, as they did not fall into any of the other clades (Figs 1B and S1). Interestingly, the alignment revealed independent gains and losses of molecular traits during plant YTHDF evolution. For example, an aspartate-to-asparagine substitution in helix α1 that increases affinity for m⁶A [61], characteristic of DC and plant DF-B YTH domain proteins [26, 61], is also present in the fern proteins of the DF-F clade, but absent in DF-Es, DF-As and gymnosperm DF-Fs (S4 Fig), suggesting independent losses or acquisitions. Taking these observations together, we conclude that land plant YTHDFs are phylogenetically more diverse than previously appreciated.

### YTHDF protein diversification occurred early during land plant evolution

Our phylogenetic analysis shows that plant YTHDF diversification started at least before the radiation of Euphyllophytes (plants with true leaves comprising ferns, gymnosperms and angiosperms) more than 410 Mya [59,63]. This is because bryophytes possess one or two

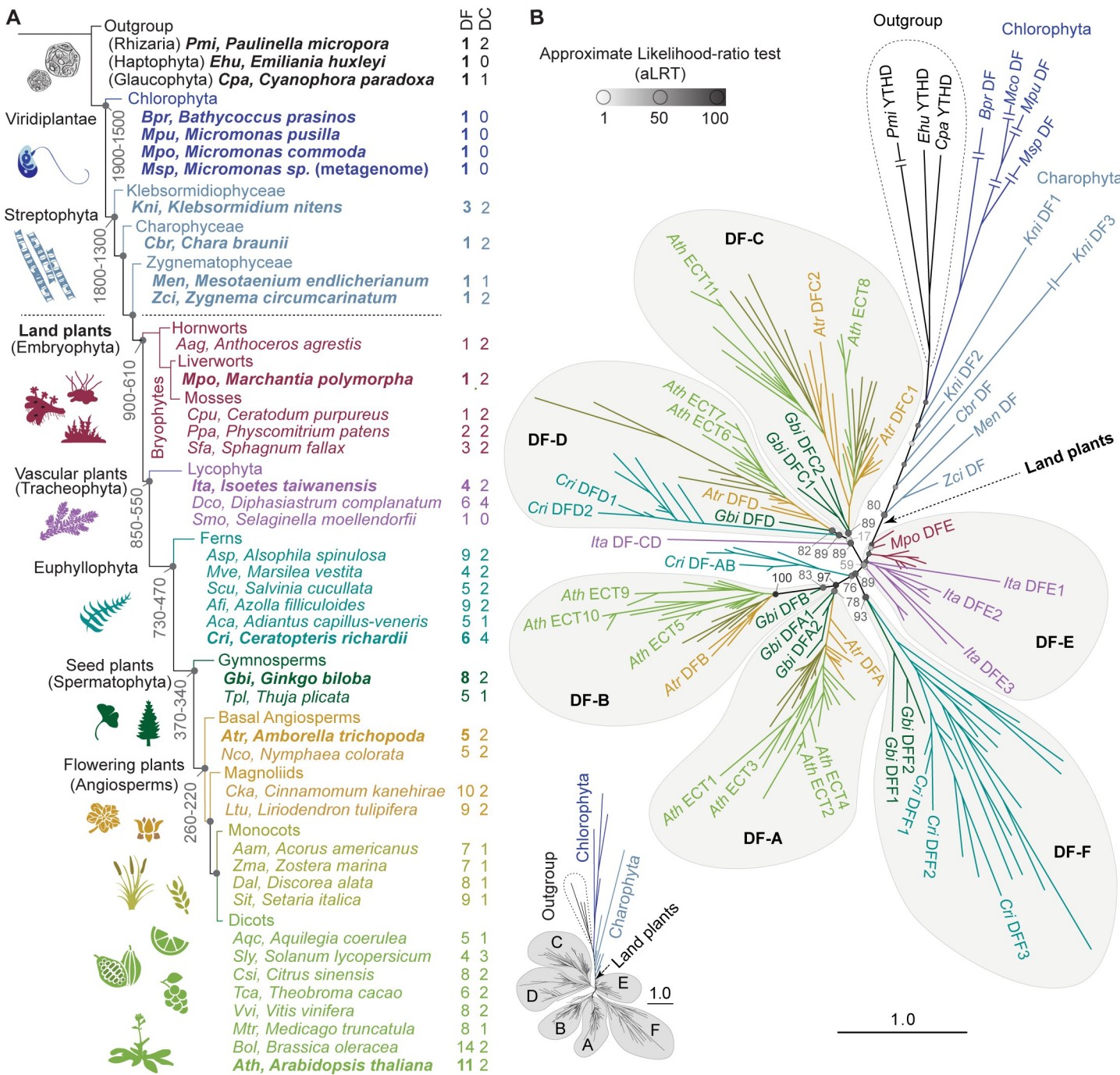

**Fig 1. Phylogenetic analysis of YTHDF proteins in plants. (A)** Schematic representation of Viridiplantae evolution and relative position of the species used in this study. The architecture of the diagram and the age (million years, My) indicated on some nodes are a simplified version of the trees from Su et al. [59] and Bowman [62]. The length of the branches has been adjusted for illustrative purposes. **(B)** Phylogenetic tree of YTHDF proteins in plants (S1–S4 Datasets), with color-coding and abbreviations of species names as in A. Three YTHDF proteins from taxa outside but evolutionarily close to Viridiplantae are included as outgroup. *Arabidopsis thaliana* (*Ath*) YTHDF proteins are named after the original nomenclature for proteins containing an Evolutionarily Conserved C-Terminal Region (ECT) established by Ok et al. [40]. Proteins from other plant species adhere to the nomenclature established by Scutenaire et al. [26], with small variations reflecting additional clades (DF-E, -F) and the split of the original DF-C clade into -C and -D. Statistical support is calculated as approximate likelihood-ratio test (aLRT), and indicated by grayscale-coded spheres and values on the most relevant nodes. For simplicity, only a subset of proteins is labelled–those from one representative species of the main taxa highlighted in bold in A–, but a comprehensive representation of the same tree with all protein names and aLRT values is available in S1 Fig. The length of the branches represents the evolutionary distance in number of amino acid substitutions per site relative to the scale bar. Double lines crossing the longest branches indicate that the branch has been collapsed due to space problems, but a schematic representation of their relative lengths is shown on the lower-left corner, and not-collapsed branches can be found in S1 Fig.

YTHDFs in the 'early clade' (DF-E), while all six fern species analyzed here have extended sets of DF proteins (4–9 paralogs) that spread into clades DF-F, DF-D, and the branch sustaining the DF-A and DF-B groups (Fig 1B). Furthermore, because one DF-protein from the lycophyte *Isoetes taiwanensis* branches from the common stem between clades DF-C and DF-D (*Ita* DF-CD), it is likely that a first diversification event started in the ancestral vascular plants. Thus, our analysis reveals that YTHDF radiation started early in land plant evolution and coincided with the acquisition of morphological complexity and the adaptation to diverse environments.

## A functional complementation assay for plant YTHDF proteins

To address the degree of functional specialization among plant YTHDF proteins in a simple manner, we set up a functional complementation assay that scores the ability of each of the eleven arabidopsis YTHDF proteins (Fig 2A) to perform the molecular functions of ECT2/3/4 required for rapid cellular proliferation in leaf primordia. Initial attempts at using the *ECT2* promoter for ectopic *ECT* expression in primordial cells were unsuccessful, as pilot experiments with a genomic *ECT4* fragment revealed that its expression was substantially lower than that of a similar *ECT2* genomic fragment when driven by the *ECT2* promoter (S5 Fig), perhaps pointing to the presence of internal *cis*-regulatory elements. We therefore turned to the use of *ECT* cDNAs (cECTs) under the control of the promoter of the ribosomal protein-encoding gene *RPS5A/uS7B/US7Y* (At3g11940 [64,65]), henceforth *US7Yp*, active in dividing cells [66] in a pattern that resembles the *ECT2/3/4* expression domain [16,17]. Transformation of the *US7Yp:cECT2-mCherry* construct in *te234* plants resulted in complementation frequencies of ~40–50% among primary transformants, a percentage slightly lower but comparable to that obtained with the genomic *ECT2-mCherry* construct under the control of *ECT2* promoter and terminator regions (*ECT2p:gECT2-mCherry*) [16] (S6A Fig). Free mCherry expressed from a *US7Yp:mCherry* transgene showed no complementation (S6A Fig), as expected. Importantly, the leaf morphology and the pattern of mCherry fluorescence in *US7Yp:cECT2-mCherry* lines was indistinguishable from that of *ECT2p:gECT2-mCherry* (S6B Fig). Hence, we proceeded with expression of cDNAs encoding all of ECT1-11 fused to C-terminal mCherry under the control of the *US7Y* promoter (henceforth ECT1-11) in *te234* mutants (Fig 2B).

## Percentage of complementation among primary transformants as a readout for functionality

To design the experimental approach in detail, we considered the fact that expression of transgenes involves severe variations in levels and patterns among transformants. This is due to positional effects of the T-DNA insertion and the propensity of transgenes to trigger silencing in plants [67], and explains why only a fraction of *ECT2-mCherry* lines rescues loss of ECT2 function (S6A Fig, [16]). Hence, many independent lines need to be analyzed before choosing stable and representative lines for further studies. Taking this into account, we decided to perform systematic and unbiased comparisons of ECT functionality by counting the fraction of primary transformants (henceforth T1s) able to complement the late leaf emergence of *te234* plants for each construct. We set a size threshold for the rapid assessment of large numbers of transformants considering that leaves longer than 0.5 mm after 10 days of growth constitute unambiguous sign of complementation, because this is the minimum size never reached by *te234* or control lines (S2B Fig). Hence, we calculated complementation percentages as the number of T1s with leaf size (s) larger than 0.5 mm relative to the total number of transformants, and averaged that score over several independent transformations (Figs 2C and S7A, S8A).

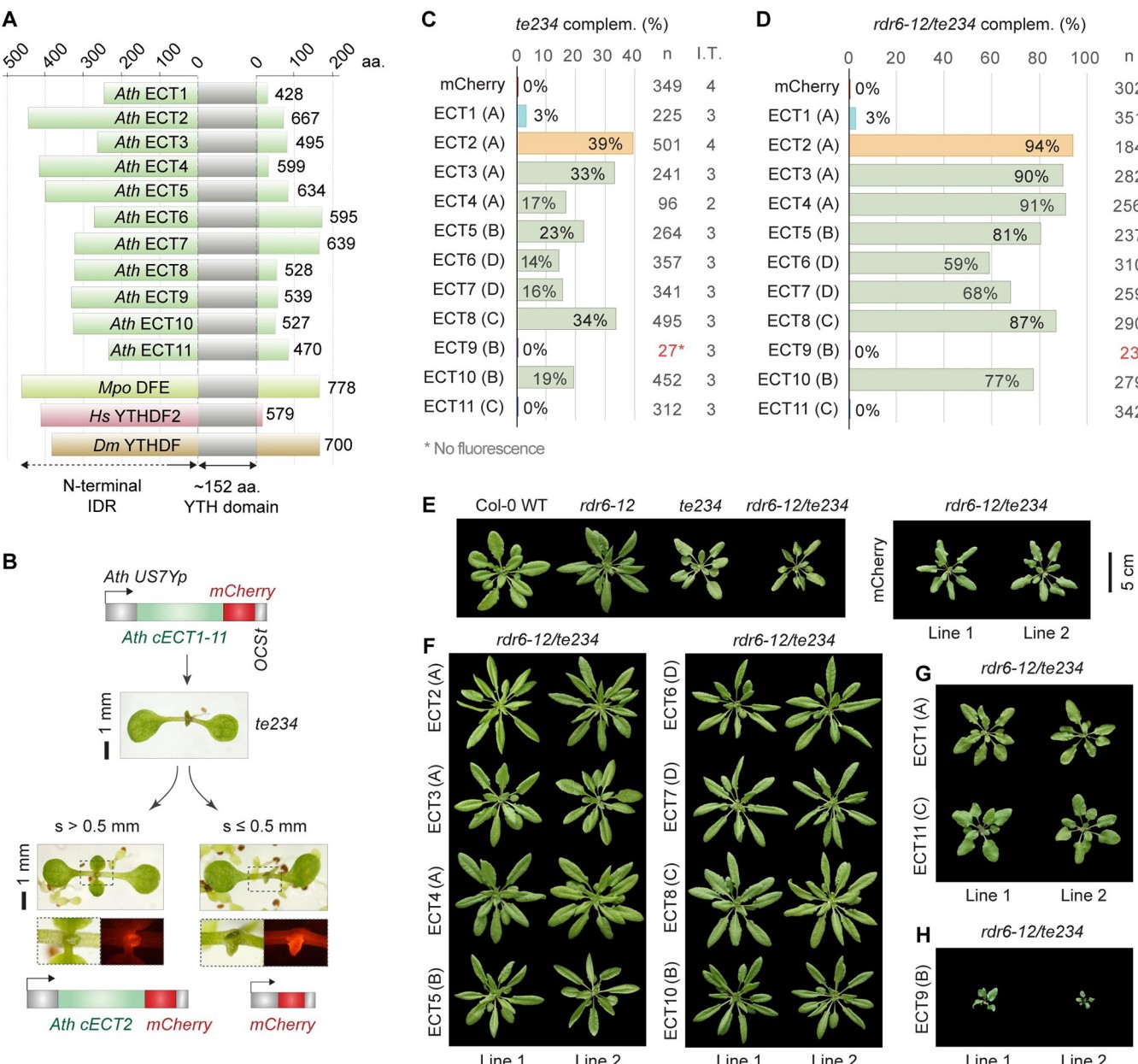

**Fig 2. Most but not all arabidopsis YTHDF proteins can replace ECT2/3/4-function. (A)** Diagram showing the relative length of the N-terminal IDRs of *Arabidopsis thaliana* (*Ath*) YTHDF proteins (ECT1-ECT11) together with *Marchantia polymorpha* (*Mpo*) DFE, *Homo sapiens* (*Hs*) YTHDF2, and *Drosophila melanogaster* (*Dm*) YTHDF. Numbers indicate the length of the proteins in amino acids (aa). **(B)** Strategy followed for the functional assay. *US7Yp:cECT(X)-mCherry* constructs are introduced in *ect2-1/ect3-1/ect4-2* (*te234*) plants, and complementation rates are estimated by the percentage of primary transformants (T1) whose first true leaves have a size (s) exceeding 0.5 mm after 10 days of growth. The *construct US7Yp:mCherry* is used as negative control. Examples of T1 seedlings expressing control and ECT2 constructs are shown. Dashed outlines are magnified below to show mCherry fluorescence in emerging leaves. **(C)** Weighed averages of the complementation percentages observed for each *US7Yp:cECT(X)-mCherry* construct in 2–5 independent transformations (I.T.). Letters between parenthesis indicate the DF- clade assigned to each protein. Colouring highlights ECT2 (reference) and proteins with low or negligible complementation capacity (ECT1/9/11) for fast identification along the manuscript. Low transformation efficiency (T.E.) for ECT9 can be inferred by the low number of transformants (n) highlighted in red, but detailed T.E. and raw complementation rates in each transformation can be found in S9 and S7A Figs respectively. Additional data regarding this assay is shown in S8 Fig. **(D)** Same as C on a single transformation using the *rdr6-12/te234* background. **(E-H)** Rosette phenotype of 32-day-old T1s bearing *US7Yp:cECT(X)-mCherry* constructs in the *rdr6-12/te234* background, sorted according to their degree of complementation in three panels: F (high), G (residual) and H (non-complementation and enhancement). Different genetic backgrounds and control transformants (mCherry) are shown in E as a reference. The scalebar (5 cm) applies to all plants in the four panels.

## Most arabidopsis YTHDF paralogs can perform the molecular function required for leaf development

The results of the comparative analysis reveal a high degree of functional overlap within the arabidopsis YTHDF family, because eight out of the eleven YTHDF proteins, namely ECT2/3/4/5/6/7/8/10, complement the delayed leaf emergence of *ect2/3/4* mutants with frequencies ranging from 14% (ECT6) to 39% (ECT2) when expressed in the same cells (Fig 2C). Importantly, complementation is indirect proof of m$^6$A-binding ability, because the proteins must bind to the targets whose m$^6$A-dependent regulation is necessary for correct leaf organogenesis in order to restore ECT2/3/4 function [16]. This conclusion is not trivial, as some eukaryotes lacking canonical m$^6$A-methyltransferases encode YTH domain proteins [68] and, indeed, the YTH domain of fission yeast Mmi1 does not bind to m$^6$A [69]. Thus, most, but not all, arabidopsis YTHDF proteins retain the molecular functions required to stimulate proliferation of primed stem cells. Of the three remaining proteins, ECT1 showed only residual complementation capacity, while no complementation was observed for ECT11 at this stage (Fig 2C). Finally, we could not conclude at this point whether ECT9 could replace ECT2/3/4 function or not, because this transgene produced very few transformants in the *te234* background with barely visible fluorescence (Figs 2C and S7–S9). All such transformants underwent complete silencing in the following generation (T2).

## Differences in complementation scores are not explained by a bias in expression levels

Our T1 scoring method relies on the assumption that the *US7Y* promoter drives similar ranges of expression of the different *ECT* cDNAs. However, different silencing propensities between the *ECT* cDNAs might introduce bias in the results. To take this possibility into account, we first performed western blotting from three lines per transgene selected as the best-complementing in their group (S8B Fig). The results showed that the higher complementation scores of e.g. ECT2, ECT3 and ECT8 are not explained by higher expression. Rather, the reverse was true. For example, while ECT2 could complement *te234* plants with the lowest expression among all ECTs, the few lines expressing ECT1 able to pass the complementation threshold had the highest expression of all ECTs, suggesting that ECT1 can only partially compensate for loss of ECT2/3/4 when overexpressed.

## The same functional classes of ECTs emerge from complementation of silencing-deficient *te234*

Next, we tested whether the same results could be reproduced in plants lacking *RNA-DEPENDENT RNA POLYMERASE 6* (*RDR6*), a gene essential for the biogenesis of transgene-derived short interfering RNAs (siRNAs) [70, 71]. Because siRNA-guided transgene silencing is impaired in *rdr6* mutants, higher and more homogenous ranges of expression among different transgenes can be expected in this background. Hence, we constructed *rdr6-12/te234* mutants that exhibited both the narrow, downward-curling leaf phenotype of *rdr6* mutants [72] and the characteristic rosette morphology of *te234* [16] (S10A Fig). Importantly, the delay in leaf emergence was identical in *te234* and *rdr6-12/te234* mutants (S10A Fig) and the complementation rates of *ECT2* transgenes were much higher in the silencing-deficient plants (S6A Fig), indicating that the <100% complementation frequencies in *te234* were indeed due to RDR6-dependent silencing. In the *rdr6/te234* background, complementation scores of the eight functionally equivalent ECTs (ECT2/3/4/5/6/7/8/10) ranged from 59% (ECT6) to 94% (ECT2), and the differences between these ECTs were less pronounced, but followed a trend

largely unchanged compared to the *te234* background (Fig 2C and 2D). On the other hand, the difference between the eight ECTs with ECT2/3/4-like function and the group of ECTs with negligible complementation capacity (ECT1/9/11) was highlighted by the *rdr6* mutation: ECT1 and ECT11 did not increase their scores (Fig 2D), and absence of silencing resulted in a number of *ECT9* transformants with visible mCherry fluorescence, yet no complementation activity (Figs 2D and S10B), thus verifying the lack of ECT2/3/4-like function of ECT9.

## Rosette phenotypes corroborate the complementation scores

To verify the results of the complementation assay in the adult phenotype, we characterized rosettes of two independent lines for each transgene (Fig 2E–2H). As expected, *rdr6-12/te234* plants expressing ECTs with high complementation scores resembled single *rdr6-12* mutants (Fig 2E and 2F), while the ECT1 and ECT11 transformants with the biggest first true leaves at 10 DAG produced plants only slightly bigger than the *rdr6-12/te234* background (Fig 2E and 2G), indicating low but residual ECT2/3/4-like function for these two proteins. Strikingly, *rdr6-12/te234* plants expressing ECT9 exhibited strong developmental phenotypes (Figs 2H and S10B) that resembled those of severely m$^6$A-depleted plants [51,52]. Such an effect was not observed for any other *Ath* ECT. This result suggests that ECT9 can bind to m$^6$A targets, either exerting opposite molecular actions compared to ECT2/3/4, or simply precluding their possible regulation by remaining ECTs such as ECT5/6/7/8/10 by competitive binding. We tested whether ECT9 can outcompete endogenous ECT2/3/4 in the proliferating cells of the apex by expressing the ECT9 transgene in single *rdr6-12* mutant plants. We found no evidence for this, because the transformation produced many fluorescent T1s without obvious defects (S11 Fig). Altogether, our results indicate that arabidopsis YTHDF proteins may be divided into three functional classes, i) ECT2/3/4/5/6/7/8/10 with the molecular functionality of ECT2/3/4 required for correct and timely leaf formation; ii) ECT1/11, with only residual ECT2/3/4 functions, and iii) ECT9, unable to perform such functions and toxic when present in the expression domain of ECT2/3/4 in their absence.

## Validation of the functional assay by loss-of-function genetic analysis

One of the predictions on genetic interactions between *Ath ECT* genes that emerges from our results is particularly unexpected and, therefore, suitable for the validation of our complementation assay by classical genetic analysis: Mutation of *ECT1* should not exacerbate the developmental defects of *te234*, even if it is endogenously expressed in leaf primordia. *ECT1* lends itself well to validation for two reasons. First, ECT1 is the closest paralog of ECT2/3/4 (Fig 1B), intuitively suggesting some degree of genetic redundancy. Second, *ECT1* promoter-reporter fusions [40] and mRNA-seq data ([73], S12 Fig) suggest that expression levels and tissue specificity of *ECT1* are comparable to that of *ECT4*, whose activity is easily revealed as an enhancer mutant of *ect2/ect3* [16].

## ECT1 and ECT2/3/4 are cytoplasmic and are expressed in the same cell types

We first assessed the expression pattern and subcellular localization of ECT1 using stable transgenic lines expressing TFP translationally fused to the C-terminus of *ECT1* (*ECT1-TFP*, S13A Fig). The pattern of turquoise fluorescence was strongly reminiscent of that of fluorescent ECT2/3/4 fusions [16] with signal at the base of young leaves, in leaf primordia, main root tips and lateral root primordia at different stages (Fig 3A–3D). In addition, confocal microscopy of meristematic root cells showed cytoplasmic ECT1-TFP signal with heterogenous texture similar to ECT2/3/4 [16] (Fig 3E). We noticed, however, that most root tips had a few

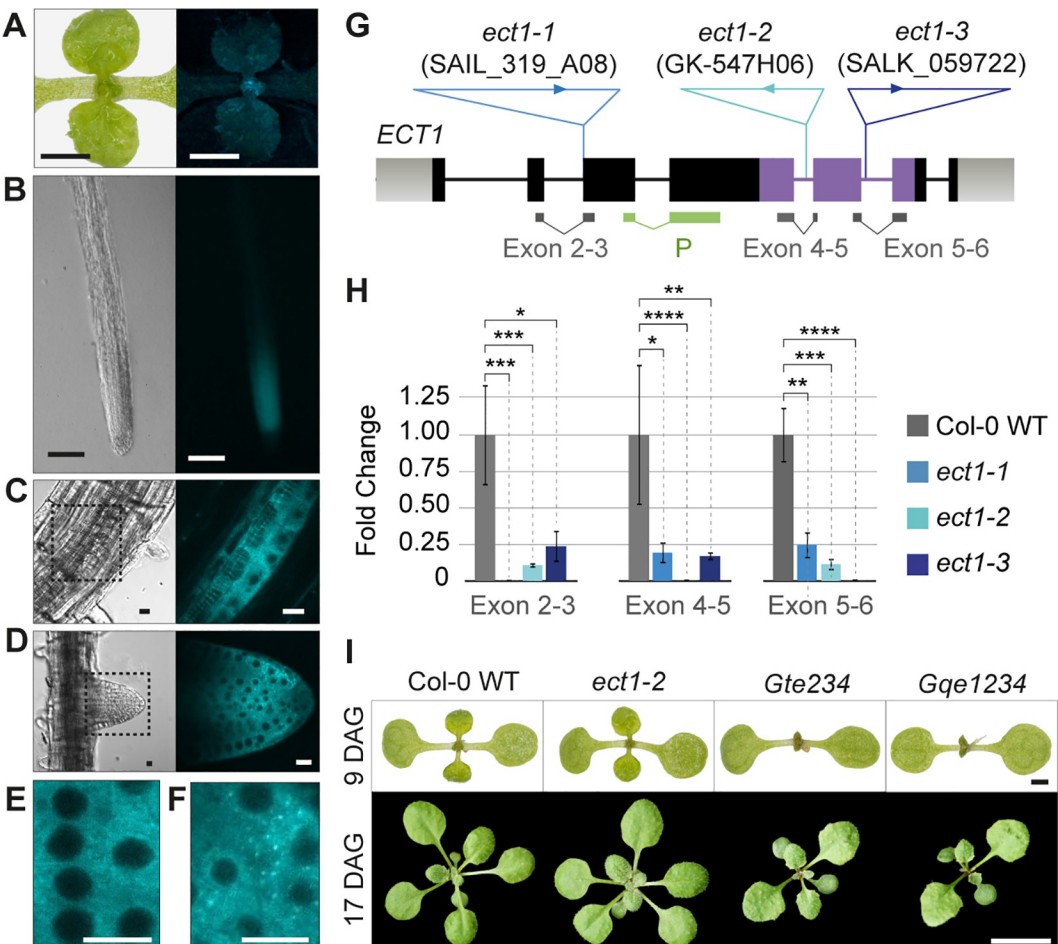

**Fig 3. Endogenous ECT1 has negligible ECT2/3/4-like function, despite similar expression pattern and subcellular localization. (A-D)** Expression pattern of *ECT1p:gECT1-TFP* (gDNA) in aerial organs (A), main root (B), lateral root primordia (C) and emerging lateral roots (D) of 10-day-old seedlings. The expression mimics the pattern observed for ECT2/3/4 (Arribas-Hernández et al., 2020). **(E-F)** Intracellular localization of ECT1-TFP in meristematic cells of root tips in unchallenged conditions. Although the cytoplasmic signal is largely homogenous (E), sporadic foci (F) are frequently observed. S13A Fig shows the integrity of the fluorescently tagged protein in independent transgenic lines assessed by protein blot. **(G)** Schematic representation of the *ECT1* locus. Exons are represented as boxes and introns as lines. The positions and identifiers of the T-DNA insertions assigned to the *ect1-1*, *ect1-2* and *ect1-3* alleles are marked, and the location of qPCR amplicons and hybridization probes (P) for analyses is indicated below. Genotyping of *ect* alleles is shown in S13C–S13F Fig. **(H)** Expression analysis of *ECT1* mRNA in wild type and T-DNA insertion lines by qPCR. *P<0.01, **P<0.001, ***P<0.0001, and ****P<0.00001 in T-test pairwise comparisons (supporting data in S9 Dataset). Northern blot using the probe (P) marked in G detects *ECT1-TFP* mRNA, but the endogenous *ECT1* transcript is below detection limit (S13B Fig). **(I)** Morphological appearance of seedlings with or without *ECT1* in the different backgrounds indicated. Nomenclature and additional phenotyping of these and alternative allele combinations can be found in S14 Fig. DAG, days after germination. Scale bars are: 1 cm in A, 1 mm in B, 100 µm in C, 10 µm in D-F, 1 mm in upper panels of I, and 1 cm in the lower panels.

cells containing distinct ECT1-TFP foci (Fig 3F) in the absence of stress. We found no trace of free TFP that could be causing imaging artifacts (S13A Fig) but note that the *ECT1-TFP* transgenic lines selected on the basis of visible and reproducible pattern of fluorescence may overexpress *ECT1* (S13B Fig). We conclude from the similar expression pattern and subcellular localization of ECT1 and ECT2/3/4 that assessment of the possible exacerbation of *te234* phenotypes by additional loss of *ECT1* function is a meaningful test of our functional assay.

## Mutation of *ECT1* does not exacerbate the phenotype of *ect2/ect3/ect4* plants

We isolated three homozygous knockout lines caused by T-DNA insertions in the *ECT1* gene, *ect1-1*, *ect1-2*, and *ect1-3* (Figs 3G and 3H and S13C–S13F), and crossed two of them to different *ect2/ect3/ect4* triple mutants [16,17] to obtain two independent allele combinations of quadruple *ect* mutants, *qe1234* and *Gqe1234* (see S14A Fig for nomenclature). In both cases, the quadruple mutant plants were indistinguishable from the corresponding *te234* and *Gte234* triple mutant parentals (Figs 3I and S14B). Similarly, triple *ect1/ect2/ect3* mutants were identical to double *ect2/ect3* mutants, and single *ect1* mutants did not show any obvious defects in *ect2/3*-associated traits such as leaf or root organogenesis and trichome branching (Figs 3I and S14B–S14D). Thus, despite their similar expression pattern, there is no indication of redundancy between ECT1 and the other three *Ath* DF-A paralogs. This unexpected conclusion agrees with the poor ECT2/3/4-like activity of ECT1 in our functional assay, thereby validating the approach.

## The divergent function of ECT1 and ECT11 is caused mainly by properties of their IDRs

Having established that different molecular functions are represented in the arabidopsis YTHDF family, we went on to map the molecular regions responsible for this functional divergence. First, we investigated whether lack of *te234* complementation by ECT1/9/11 is due to altered target binding by the YTH domain, IDR-related effector functions, or a combination of both. Thus, we built chimeric IDR/YTH constructs between ECT2 and ECT1/9/11 (Fig 4A).

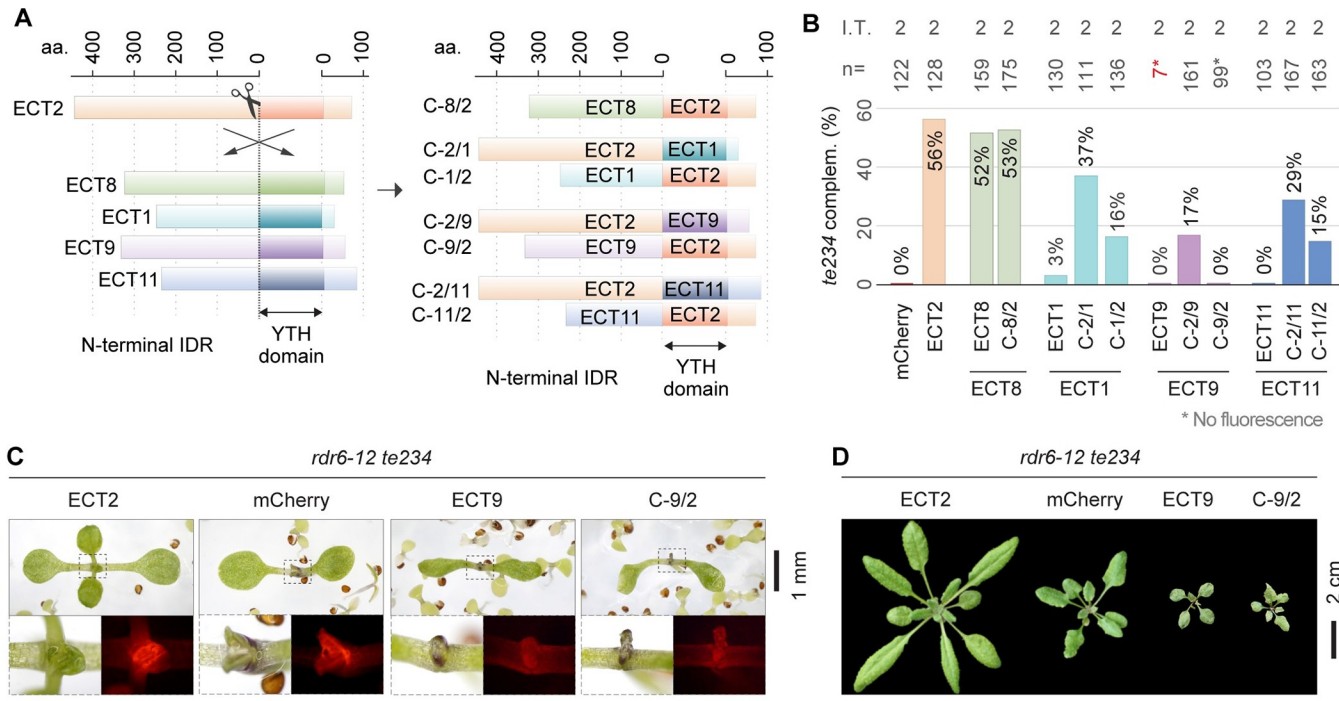

**Fig 4. Dissection of ECT1/9/11 functional regions. (A)** Schematic representation of the strategy followed to express chimeras with N-terminal IDR and YTH domains of different ECT proteins. **(B)** Weighed averages of the complementation rates observed for each chimeric construct measured as in Fig 2B and 2C. The raw complementation rates and transformation efficiencies of each independent transformation (I.T.) can be found in S7B and S9 Figs respectively, and photographs of the scored 10-day-old T1 seedlings with mCherry fluorescence are shown in S15 Fig. **(C)** 10-day-old primary transformants of *rdr6-12/te234* with the indicated transgenes. Dashed outlines are magnified below to show mCherry fluorescence. **(D)** Same genotypes as in C at 26 days after germination. Additional independent transgenic lines, developmental stages and controls for C and D can be found in S10 Fig. All transgenes are expressed from the *US7Y* promoter and contain mCherry fused to the C-terminus.

We also tested an ECT8$_{IDR}$/ECT2$_{YTH}$ chimera (C-8/2) to assess whether hybrid constructs can at all be functional, choosing ECT8 as a positive control due to its robust complementation of seedling and adult phenotypes (Fig 2C–2D, 2F). Expression of the constructs in *te234* mutants showed that the C-8/2 chimera was fully functional (Figs 4B and S7B, S15), thus validating the approach. The ECT1- and ECT11-derived hybrids showed that their YTH domains retain m$^6$A-binding ability, because the complementation frequencies for the C-2/1 and C-2/11 chimeras were not zero (Fig 4B), contrary to an m$^6$A-binding deficient point mutant of *ECT2* [16]. They also proved, however, that both the IDR and the YTH domain of these two proteins only retain a reduced degree of function compared to the equivalent regions in ECT2. In particular, the IDRs of both proteins scored lower than their YTH domains when combined with the other half from ECT2 (Fig 4B), pointing to divergence in the IDR as the main cause for different functionality. In wild type ECT1 and ECT11 proteins, the combination of the two poorly-performing parts is likely the cause of the minute ECT2/3/4-like activity.

### The IDR of ECT9 is incapable of performing ECT2/3/4 molecular functions, but its YTH domain retains some ECT2/3/4-like function

The ECT2$_{IDR}$/ECT9$_{YTH}$ chimera (C-2/9) yielded normal transformation efficiencies and up to 17% of T1s had signs of complementation compared to 56% for ECT2 (Figs 4B and S7B, S9, S15), suggesting that the YTH domain of ECT9 retains at least partial functionality, including m$^6$A-binding activity. However, the reverse construct (ECT9$_{IDR}$/ECT2$_{YTH}$, C-9/2) recapitulated the results obtained for ECT9: low transformation efficiency and low fluorescence in the few T1s recovered, and complete silencing in the next generation (Figs 4B and S7B, S9, S15). Hence, we introduced the C-9/2 chimera into the *rdr6-12/te234* background and observed plants with stronger fluorescence exhibiting aberrant morphology, again similar to what was obtained with ECT9 (Figs 4C, 4D and S7C, S10B). These results confirm that ECT9 is not functionally equivalent to ECT2/3/4, and point to the IDR as the main site of functional divergence of the protein.

### The molecular requirements for m$^6$A-binding are conserved in the YTH domains of ECT1/9/11

To pinpoint molecular characteristics of ECT1/9/11 responsible for their different functionality, we performed a series of comparative analyses between ECT1/9/11 and other ECT proteins. We first analyzed the YTH domains, taking ECT2 as a reference and using the structurally well-characterized YTH domain of *Homo sapiens* (*Hs*) YTHDF1 in complex with RNA GGm$^6$ACU [61] as a model. Amino acid sequence alignment of the YTH domains of all *Ath* ECTs (S16 Fig), Alpha-Fold-predictions [74, 75] of the apo structures and homology models [76] of the m$^6$A-bound state (S17 Fig) confirmed a high degree of conservation of the key residues required for m$^6$A-RNA binding and overall fold of the YTH domain of ECT1/9/11, in agreement with the partial functionality observed in our chimeric constructs (Fig 4B). However, we also detected a few non-conservative amino acid substitutions specific for ECT1/9/11 (Figs 5A and S16), preferentially located on the surface of the protein but far from the RNA-binding groove (S17 Fig). These non-conservative substitutions are prime candidates to explain the minor differences in functionality, a scenario that implies biological functions of YTH domains other than mere m$^6$A-binding capacity.

### The IDRs of ECT1/9/11 have biophysical properties distinct from ECT2/3/4/5/6/7/8/10

We next focused on the N-terminal regions of *Ath* ECT1/9/11. Because protein disorder predictions using MobiDB [77] confirmed that they are all disordered (S18 Fig), we used

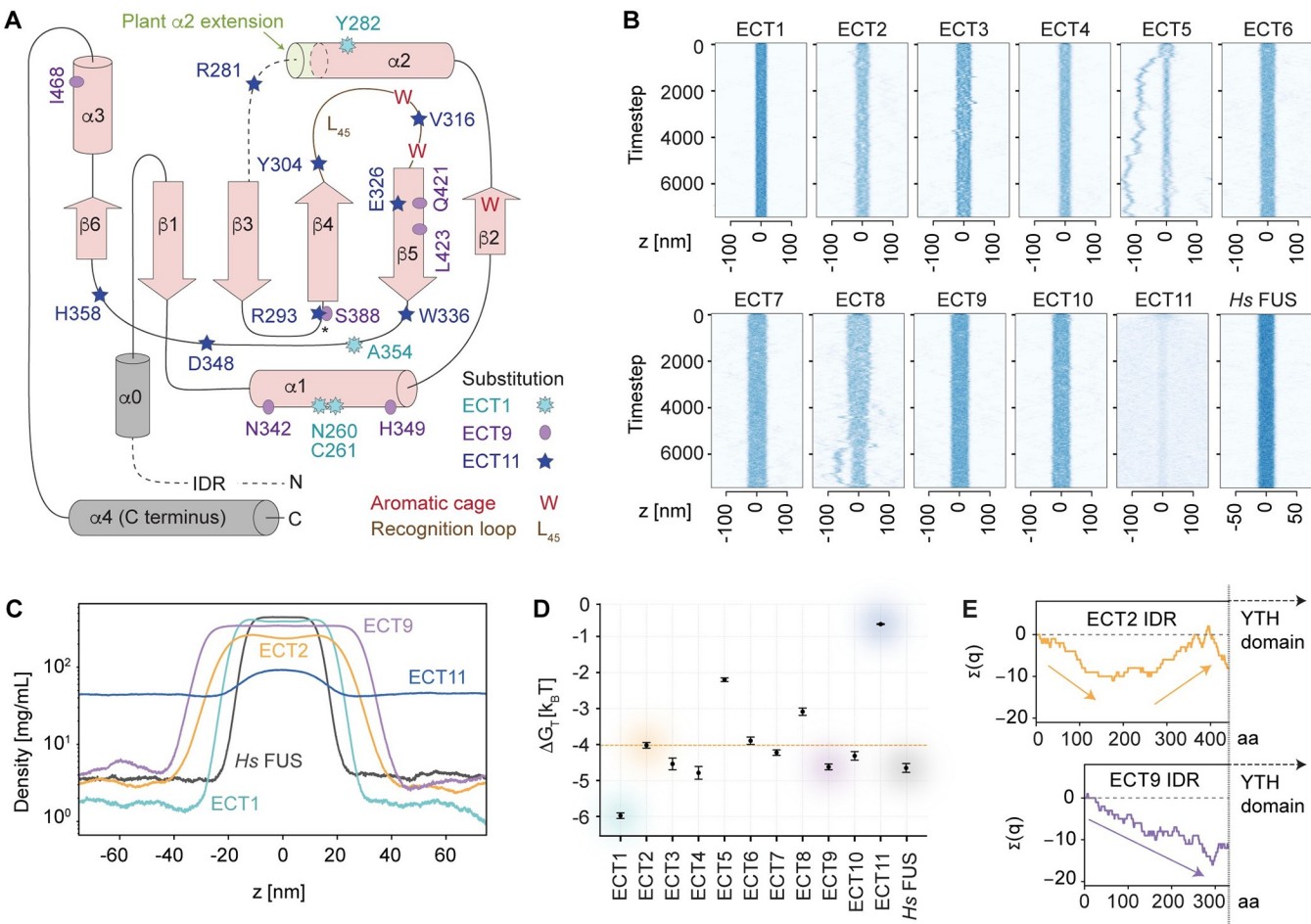

**Fig 5. Analysis of ECT1/9/11 functional regions. (A)** Topology of the YTH domain of *Homo sapiens* (*Hs*) YTHDF1 with secondary structure elements drawn in pink [61]. The two grey helices represent the only structural elements outside the YTH domain as defined by Stoilov et al. [8]: α0 immediately upstream, and α4 at the C-terminus of the protein. Superimposed on the diagrams are the non-conservative amino acid substitutions found in *Ath* ECT1, ECT9 and ECT11 compared to the majority of *Ath* ECTs (see alignment in S16 Fig). The localization of these amino acids in the predicted 3D structures, predominantly on the surface, and the spatial integrity of the aromatic cage in the homology models of ECT1/9/11, can be found in S17 Fig. **(B)** Slab simulations of the IDRs of the indicated proteins showing their tendency to remain in condensed phase or disperse in solution. Source videos of ECT2, ECT1 and ECT11 simulations are provided as S1–S3 Movies **(C)** Time-averaged density profiles for a subset of the proteins in B. The analyses of the remaining *Ath* ECTs are shown in S19A Fig. **(D)** Calculated free energy change for the transition between aqueous solution and condensed phase ($\Delta G_{trans}$ = RT ln [$c_{dilute}$ / $c_{dense}$]) for the IDRs of the indicated proteins. **(E)** Cumulative charge distribution of the IDRs of the indicated proteins. The length in amino acids (aa) is at scale. The same analysis for all *Ath* ECTs can be found in S19D Fig.

approaches suitable for the study of IDRs. Prediction of short linear motifs (SLiMs) that may mediate interaction with effector proteins gave many possibilities that did not follow a clear pattern when compared between the different ECT classes (S18 Fig). Hence, we decided to study other properties known to be important in IDRs and focused on the propensity to form liquid condensates, the amino acid composition, and the distribution of charge. To determine whether the distinct functional classes of ECTs differed in their propensity to phase separate, we employed a recently developed coarse-grained model of IDRs with residue-specific parameters estimated from experimental observations of many IDRs [78] to simulate their tendency to remain in the condensed phase or disperse in solution. As control, we used the well-studied IDR of the human RNA-binding protein Fused in Sarcoma (FUS) [79–82]. The results showed that the IDR of ECT2 has a clear propensity to remain in the condensed phase (Fig 5B–5C and

S1 Movie) with a highly favorable Gibb's free energy change for the transition from dilute aqueous solution to condensed phase (Fig 5D), comparable to, albeit not as pronounced as for the IDR of *Hs* FUS (Fig 5B–5D). While the IDRs of most *Ath* ECT proteins behaved comparably to ECT2, the IDRs of ECT1 and ECT11 were clear outliers (Figs 5B–5D and S19A). Compared to all other ECTs, the IDR of ECT1 showed a much stronger propensity to phase separate, consistent with the spontaneous granule formation of ECT1-TFP *in vivo* (Figs 3F and 5B–5D, and S2 Movie). On the contrary, the IDR of ECT11 dispersed into solution much more readily (Fig 5B–5D and S3 Movie). Thus, the IDRs of ECT1 and ECT11 have markedly different biophysical properties from those of ECT2/3/4/5/6/7/8/10, perhaps contributing to their different functionality *in vivo*. In contrast, the IDR of ECT9 did not stand out from the rest in the simulations of phase separation propensity. We also note that the different phase separation propensity among the ECTs is driven, in the main, by their different content of sticky residues as defined by the CALVADOS model [78] (S19B Fig). In this context, it is potentially of interest that ECT1, ECT9 and ECT11 all have a higher content of Phe residues than the nearly Phe-depleted IDRs of ECT2/3/4/5/6/7/8/10 (S19C Fig). For ECT1, whose Tyr and Trp enrichment is comparable to that of ECT2/3/4/5/6/7/8/10 (S19C Fig), the additional enrichment of the similarly sticky Phe may contribute to its more pronounced phase separation propensity.

We finally analyzed the charge distribution, as this may dictate properties such as compactness and shape of IDR conformations [83, 84]. We found that the IDRs of ECT2/3/4/5/6/7/8/10 show a recurrent pattern with accumulation of net negative charge towards the N-terminus, and recovery to near-neutrality closer to the YTH domain, resulting in creation of electric dipoles (Figs 5E and S19D). The IDRs of ECT1, ECT9 and ECT11 differ from this pattern, but in distinct ways: ECT1 does not show a pronounced patterning of the charged residues, while ECT9 increasingly accumulates negative charge (Fig 5E) and ECT11 increasingly accumulates positive charge (S19D Fig). It is an interesting possibility, therefore, that charge separation in the IDR is a requirement to fulfill the molecular functions of ECTs to accelerate growth in primordial cells.

## The ability to complement ect2/ect3/ect4 does not simply follow affiliation with phylogenetic clades

We next compared the functional groups of YTHDF proteins defined by our complementation assays with the YTHDF phylogeny. Strikingly, the ability to rescue defective timing of leaf emergence does not follow the defined phylogenetic clades, because both high and low/null complementation capacities are present in the same clades. For example, ECT2, ECT5 and ECT8 in clades DF-A, -B and -C respectively present high ECT2/3/4-like functionality, while ECT1, ECT9 and ECT11 in these same clades do not (Figs 1B and 2C–2H). It is noteworthy, however, that there is a clear trend for non-complementors to be amongst the most highly divergent proteins in each group, as revealed by the length of the branches in the phylogenetic tree (Fig 1B). Indeed, some of the longest branches in both our tree and the one in the study by Scutenaire et al. [26] correspond to ECT1, ECT9, and ECT11. Accordingly, the three proteins have the highest number of amino acid substitutions compared to the only YTHDF protein of the liverwort *Marchantia polymorpha* (*Mpo* DFE) [85] (Fig 6A). These results suggest that the common ancestor of YTHDF proteins in land plants may have had a molecular function related to that of the modern *Ath* ECT2/3/4, and that this function has been conserved in the least-divergent members of the DF-A/B/C/D clades. Subsequent duplication and neofunctionalization may have driven the rapid evolution of highly-divergent YTHDF proteins (ECT11, ECT1, and ECT9 in arabidopsis) that have lost their primary function and perhaps fulfill different molecular roles in the plant m$^6$A pathway.

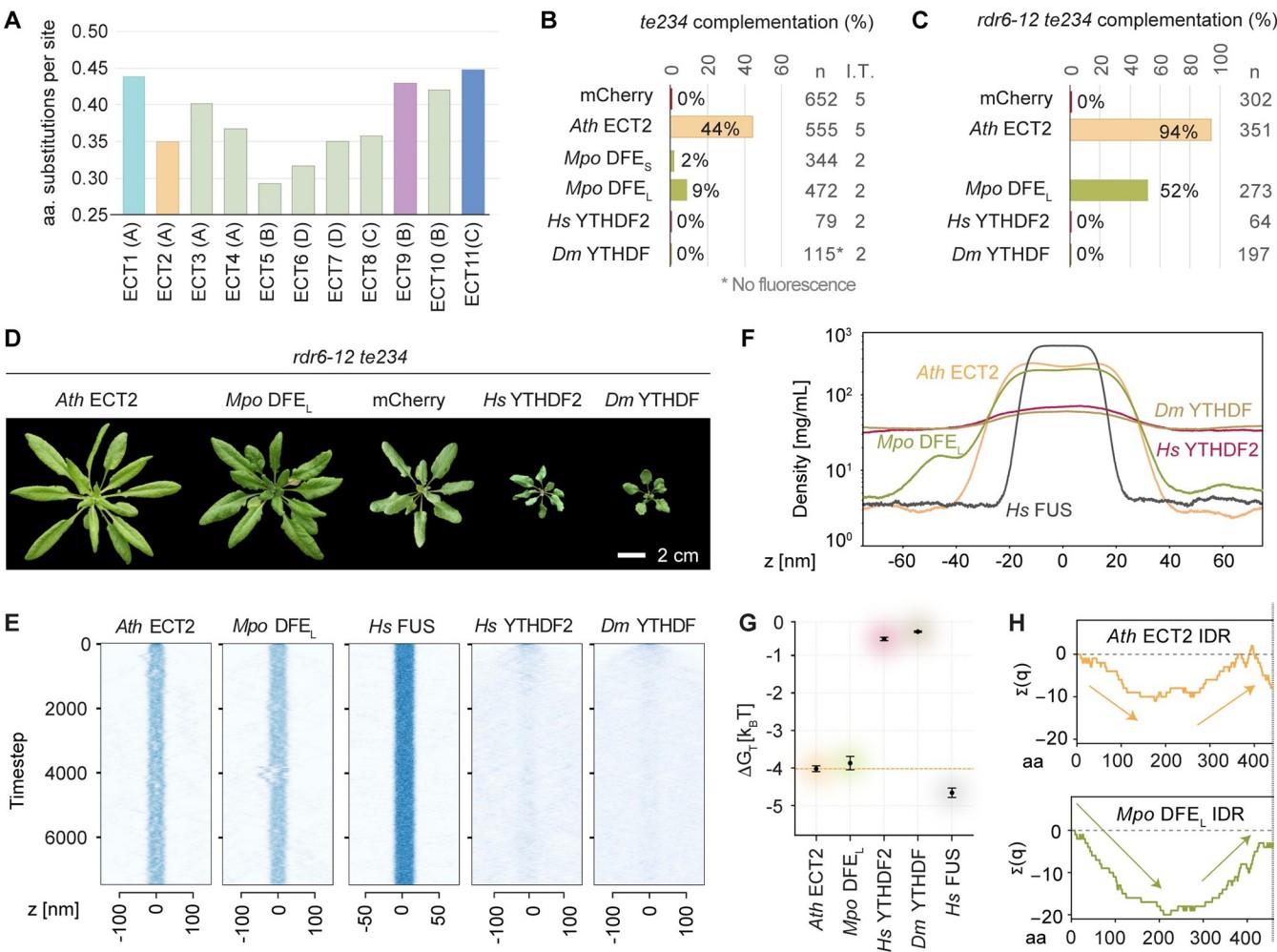

**Fig 6. Heterologous expression of a bryophyte DFE rescues loss of ECT2/3/4 function in arabidopsis, while metazoan YTHDF proteins aggravates it. (A)** Evolutionary divergence of *Arabidopsis thaliana* (*Ath*) ECT proteins estimated by pairwise comparisons between these and the only DF protein in *Marchantia polymorpha* (*Mpo* DFE) according to the alignment of YTHDFs from bryophytes and angiosperms in S7–S8 Datasets (see Methods). **(B-C)** Weighed averages of the complementation rates observed for *Mpo* DFE (S/L refer to the Short and Long splice forms found in meristematic tissues–apical notches and gemma cups–of *M. polymorpha* thalli), *Homo sapiens* (*Hs*) YTHDF2 and *Drosophila melanogaster* (*Dm*) YTHDF in the *te234* (B) or *rdr6-12/te234* (C) backgrounds, calculated as in Fig 2C–2D. n, total number of transformants; I.T. number of independent transformations (only one in C). Raw T1 counts, complementation rates and transformation efficiencies can be found in the S6 Dataset, and S7 and S9 Figs. **(D)** 32-day-old primary transformants expressing the transgenes analysed in C. Additional controls and developmental stages are shown in S10B Fig. **(E)** Slab simulations of the IDRs of the indicated proteins as in Fig 5B. **(F)** Time-averaged density profiles of the slab simulations in E. **(G)** Calculated free energy change for the IDRs of the indicated proteins as in Fig 5D. **(H)** Cumulative charge distribution of the IDRs of the indicated proteins, as in Fig 5E. The same analysis for the two metazoan YTHDFs can be found in S19D Fig.

## The molecular function of ECT2/3/4 was present in the first land plants

To test whether the common ancestor of plant YTHDF proteins indeed had a molecular function similar to that of the modern *Ath* ECT2/3/4, we subjected *Mpo* DFE (Fig 2A) to our functional assay, as bryophytes are the earliest-diverging taxon of land plants (Fig 1A, [54,56–60]). The result showed a partial, but clear capacity of two different splice forms of *Mpo* DFE to complement the delay in leaf emergence and morphology defects of arabidopsis *te234* plants (Figs 6B and S7D, S20A–S20B). Although the complementation scores were low (Fig 6B), a thorough inspection of the fluorescence intensity in all primary transformants revealed that the transgene was only expressed very weakly in a reduced subpopulation of plants, possibly due to the tendency to trigger silencing upon heterologous expression of a long cDNA from a

distant plant species [86]. We therefore introduced the best-complementing isoform into *rdr6-12/te234* mutants. This approach yielded a much higher complementation score of 52% (Fig 6C), and plants very similar to those expressing ECT2 in the same background (Figs 6D and S6B). Thus, the *Ath* ECT2/3/4 activity needed to stimulate proliferation of primed stem cells was present in the first embryophytes, suggesting that this activity is an ancestral molecular function of YTHDF proteins in land plants. Gratifyingly, the simulations of the behavior of the IDR of *Mpo* DFE showed a dipole-like charge pattern and a phase separation propensity similar to those of the ECT2/3/4/5/6/7/8/10 group, further substantiating that these IDR characteristics correlate with the ability to fulfill ECT2/3/4 function in leaf formation (Fig 6E–6H). We investigated this interesting possibility more deeply by analysing the IDRs of YTHDF proteins from species representing the main clades of land plants that diverged prior to flower evolution, a moss, a lycophyte, a fern and a gymnosperm, and from the sole species in the sister group to all flowering plants, *A. trichopoda* [87, 88]. This analysis showed that at least one YTHDF protein with ECT2-like charge distribution and phase separation propensity is present in all these taxa (S21A Fig). In particular, this included the only YTHDF protein in the moss *Ceratodum purpureus*. Interestingly, when the proteins from the different taxa were analysed together, a general trend for increased phase separation propensity that again correlated with higher 'stickiness' ($\lambda$) was apparent in proteins of clades -A, -B and -E compared to those of -C, -D and -F (S21B Fig). We also noticed that in the species that encode multiple YTHDF proteins, some paralogs exhibit divergent IDR features similar to those seen in *Ath* ECT1/9/11, but this phenomenon was not clade-specific (S21 Fig). This result suggests that the IDR-based functional diversification observed in arabidopsis may apply more generally. We conclude that the molecular function of YTHDF proteins required for stimulation of primordial cell proliferation is deeply conserved in land plants, and that the biophysical properties of the IDRs of *Mpo* DFE and the *Ath* DF proteins capable of performing this function are recurrent in all the major plant taxa.

## Heterologous expression of animal YTHDFs enhances the phenotype caused by loss of ECT2/3/4 in Arabidopsis

Finally, we subjected *Hs* YTHDF2 and *Drosophila melanogaster* (*Dm*) YTHDF (Fig 2A) to our functional assay to test whether they could function molecularly like plant YTHDFs. While we did not observe complementation of the delayed leaf emergence of *te234* plants in any of the transformants (Figs 6B and S7D), several *Hs* YTHDF2 lines showed more acute developmental defects compared to *te234* (S20B Fig). Although these defects were not observed among *Dm* YTHDF lines, no expression could be detected in these plants either (S7D Fig). Hence, we repeated the transformation in the *rdr6-12/te234* background, this time obtaining numerous transformants of both *Dm* YTHDF and *Hs* YTHDF2 with strong fluorescence (Figs 6C and S10B). These plants showed aberrant morphology comparable to the effect caused by expression of ECT9 and C-9/2, with resemblance to arabidopsis mutants with severe depletion of m⁶A [51,52] (Figs 6D and S10B). Thus, expression of metazoan YTHDFs in arabidopsis *ECT2/3/4*-deficient plants exacerbates their developmental phenotype. This result is in agreement with the generally accepted idea that m⁶A-YTHDF2 destabilizes mRNA in mammals [9,89–92] while the m⁶A/ECT2/3/4 axis does the opposite in plants [23,51,93–97]. However, an analogous molecular mechanism between plant and animal YTHDFs cannot be completely ruled out, because a different cellular context may be hampering the molecular activity of metazoan YTHDFs in plant cells, and simple competition for targets with remaining ECTs could be the cause of the enhanced phenotype. Nonetheless, we note that the IDRs of *Hs* YTHDF2 and *Dm* YTHDF had much weaker phase separation propensity (Fig 6E–6G) and different charge

distribution (S19D Fig) compared to ECT2, perhaps contributing to the lack of ECT2/3-like function *in planta*.

## Discussion

Our study combines thorough phylogenetic analyses with functional complementation to assess the extent to which molecular functions of YTHDF proteins have diverged over the course of land plant evolution. The key observation is that 8 of the 11 *Ath* ECTs and, remarkably, the sole YTHDF of the liverwort *M. polymorpha*, have the molecular functions required for leaf development in arabidopsis. Along with the refined phylogenetic relationships between plant YTHDFs that we present, these findings have implications for our understanding of three distinct, if obviously related, areas that we discuss in more detail below: (1) the molecular properties of land plant YTHDF proteins, (2) the ancestral YTHDF functions and the evolutionary paths towards YTHDF diversification, and (3) the biology underlying the recurrent employment of expanded YTHDF families in higher plants.

### 1. Inferences on molecular properties of land plant YTHDF proteins

Our study provides at least four new insights into molecular properties of land plants YTHDF proteins, all of which are important for future, more detailed mechanistic studies of eukaryotic m$^6$A-YTHDF function. First, the functional assays with *Ath* ECT proteins and the chimeras combining the YTH domain of ECT1/9/11 with the IDR of ECT2 allow us to infer that all 11 arabidopsis YTHDFs have m$^6$A-binding capacity, and suggest that some, if not all, of their *in vivo* functions involve m$^6$A binding. Second, the complementation analyses of the reverse chimeras show that the IDRs of *Ath* ECTs with divergent functions contribute substantially to their different molecular properties. This finding implies a key role of IDRs in defining the molecular activities of YTHDF proteins, and is in line with the conclusions of a recent study of mammalian YTHDFs [48], with the functional importance of the IDR of *Hs* YTHDF2 inferred from early tethering studies [9], and with the isolation of IDR-dependent interactors proposed to be important for *Hs* YTHDF2 and *Ath* ECT2 function [28–30, 96, 98]. Third, the ability to direct correct leaf formation correlates with a combination of two biophysical properties of the IDRs of plant YTHDFs: a clear propensity to phase separate, and an organization of charged residues that creates an electric dipole. Future studies are needed to address whether a causal relation underlies this correlation, and, if so, what its molecular basis is. Fourth, amino acid substitutions in YTH domains inferred to be capable of binding to m$^6$A also contribute, to a lower extent, to functional specialization. Because most affected amino acids are surface-exposed but not predicted to contact RNA, this observation indicates that YTH domains should not be seen as inert adaptors for m$^6$A association, as they may also participate in effector functions. It is an important goal of future studies to identify such functions and define how they contribute to functional diversification.

### 2. The ancestral function and evolution of YTHDF proteins

It is a key finding of our study that the sole *M. polymorpha* YTHDF protein possesses the molecular functions necessary for leaf formation in flowering plants. This remarkable result demonstrates deep functional conservation, across at least 450 My of evolution, and suggests that stimulating cell division in organ primordia may be the ancestral function of land plant YTHDFs. It is now a question of profound interest to determine how the molecular activity of the ancient YTHDF proteins evolved and diversified as the main eukaryotic linages came into existence. Our work provides several important new insights to guide the understanding of this process. In particular, since it is a valid question whether the main evolutionary event that

gave rise to the ancestral function of land plant YTHDFs occurred prior to their terrestrialization, we open this section of the discussion with considerations on how the available information on algal YTHDF proteins informs the debate. We close with the analysis of *Ath* ECT1 divergence as a case study on evolution of specialized YTHDF function.

**2a. YTHDF proteins in chlorophyte algae are either lost or highly divergent.**   The genomes of most of the chlorophyte species sequenced thus far do not contain *YTHDF* genes. Indeed, the single YTH domain protein in the model green alga *Chlamydomonas reinhardtii* belongs to the DC clade [26,68], and the same is true for all the species that we inspected in this and five additional orders (S2 Fig). We only found YTHDF-encoding genes in a subset of species in the basal group Mamiellophyceae (S2 Fig). Like land plant, metazoan and yeast YTHDF proteins, these chlorophyte YTHDFs have long N-terminal IDRs and no folded domains other than the YTH domain. Nevertheless, YTHDFs have diverged remarkably fast in Chlorophyta, as revealed by their long branches in the phylogenetic tree, well surpassing that of YTHDF proteins in streptophytes (land plants and charophyte algae) and even the microalgal species of Glaucophyta, Rhizaria and Haptophyta used as outgroup [99,100] (left bottom corner in Fig 1B, and S1 Fig). Furthermore, the amino acid substitutions and insertions found in some of the proteins affect the aromatic cage (S3 Fig), likely compromising their m⁶A-binding capacity. Hence, YTHDF proteins are poorly represented in chlorophytes and, as for the only YTH-domain protein of fission yeast Mmi1 [69], they may have lost m⁶A-related functions altogether.

**2b. YTHDF proteins are widespread and variably divergent in the sister clade of land plants.**   Another informative group of green algae is the charophytes, or streptophyte algae, photosynthetic organisms that thrive in fresh water or humid terrestrial habitats. Their ancestors split from chlorophytes over a billion years ago and gave rise to all land plants and extant charophytes, [55,62,101]. The genomes of most charophyte species sequenced thus far encode YTHDF proteins (S2 Fig) with intact aromatic cages and overall higher conservation than those of chlorophytes (S3 Fig), but some of them also differ considerably from their closest land plant relatives (Figs 1B and S1, S3), and may include additional folded and even transmembrane domains (S22 Fig).

**2c. Possible evolutionary scenarios.**   In the light of the observations regarding YTHDFs in green algae, we consider three possible scenarios for the acquisition of ECT2-like molecular functions in land plants:

1. The acquisition occurred during the terrestrialization of the common ancestor of all land plants or, at the latest, within the time window between terrestrialization and bryophyte divergence. In that case, charophyte YTHDFs should not exhibit ECT2-like functionality.

2. The acquisition occurred early during streptophyte algae evolution. We consider this the most likely scenario, because such acquisition could have served as exaptation (pre-adaptation) for the successful colonization of dry land, perhaps by contributing to the appearance of apical meristems [55,102]. The prediction in that case would be that charophytes from late-diverging clades, but not from basal groups, encode YTHDFs with ECT2-like functions.

3. The key evolutionary event occurred much earlier, perhaps upon chloroplast acquisition by the last common ancestor of Viridiplantae, Rhodophyta and Glaucophyta, i.e. the first Archaeoplastida, or even before that. We have, however, not been able to find any YTH domain protein in rhodophytes (red algae), and the only glaucophyte sequenced so far [103] contains a single YTHDF (S2 Fig). These observations, together with the scarcity and great divergence of chlorophyte YTHDFs, suggest that if the functional acquisition indeed

happened in ancestral Archaeoplastida or earlier, the function and even the proteins altogether were lost in most green and red algae, but kept in the streptophyte lineage that gave rise to land plants. Future functional studies of YTHDFs from Charophyta and Glaucophyta, and taxa close to Archaeoplastida such as Haptophyta, Rhizaria or Stramenopiles (S2 Fig, [99,100]) will be key for the dissection of the evolutionary origin of the ECT2-like molecular properties required for plant growth [104].

**2d. ECT1: A newly evolved gene in a Brassicaceae lineage as a case study.**   In addition to the fundamental question on evolution of the ancestral molecular function of land plant YTHDFs, it is also of interest to consider what characterizes the evolution of late YTHDF specialization. We examine *Ath* ECT1 as a case study in this regard. A thorough look at phylogenetic analyses from Scutenaire *et al.* [26] reveals that *ECT1* is present only in a few species of Brassicaceae such as *A. thaliana* and *Capsella rubella* (*Cru*). *Brassica oleracea* in the same family has a clear homolog of *Ath/Cru ECT3* but not of *ECT1*, other dicots encode *ECT3/1* orthologs in a separate cluster, and monocot DF-As arose from an entirely separate branch (S1 Fig, [26]). Therefore, *ECT1* is the product of a recent duplication of the *ECT3/1* gene in a small subgroup of dicots, that diverged more rapidly than any other DF-A clade member in arabidopsis (Fig 1B, [26]). This observation agrees with our overall conclusion that most YTHDFs in early angiosperms possessed *Ath* ECT2-like molecular functions, and only late events resulted in neofunctionalization of specific paralogs. We note that such rapid divergence appears to result primarily from changes in the IDR. We were able to identify two biophysical properties of the IDR of ECT1—its stronger predicted propensity to phase separate and its charge distribution —that are distinct from the ECT2/3/4/5/6/7/8/10 group, but note that additional changes, e.g. in SLiMs mediating interaction with regulatory factors, may also contribute to the different functionality [98].

## 3. Functional versus biological redundancy and the need for many YTHDFs

Our study reveals that most YTHDF proteins in higher plants can perform, essentially, the same molecular functions that allow ECT2/3/4 to promote cell proliferation in organ primordia and control organogenesis. We anticipate that this molecular function may be used in distinct biological contexts by ECT5/6/7/8/10 due to different expression patterns and response to stimuli, thereby achieving biological specialization. Such variability in expression patterns can be inferred from available RNA-Seq data in the arabidopsis eFP Browser (BAR.toronto.ca) that show, for example, that ECT8 is primarily expressed during senescence-associated processes, perhaps in connection to re-growth after reversible proliferative arrest [105,106]. Clearly, however, higher plants do not only use many YTHDF-encoding genes to provide the same molecular function(s) in distinct expression domains, as shown by the specialized molecular activity of *Ath* ECT1, ECT11, and most prominently ECT9. Whether these divergent proteins act to counteract the ECT2/3/4-like function by competing for m$^6$A-containing transcripts, perhaps halting growth upon stress, or whether they perform entirely independent functions in different tissues or cell types, is open for new investigations.

An outstanding question not yet resolved by our study is why higher plants seem to need many YTHDF proteins. Indeed, the number of paralogs in vascular plants well surpasses the three YTHDFs present in most vertebrates [6] and the even smaller sets in other eukaryotes (S2 Fig, [8,68]). The need for many YTHDFs of considerable variability is all the more puzzling given our demonstration here that the majority of them retains the same molecular functions. Robustness is a commonly given explanation for redundancy, but the same argument could apply to YTHDCs, typically encoded by 0–2 genes in most eukaryotes including land plants

[8,68] (S2 Fig), or methyltransferase subunits, for which only single copies have been maintained during evolution [68]. Therefore, the answer should probably be sought elsewhere. It is clear now that the m⁶A/YTHDF axis controls cellular proliferation and organ growth in plants [17,97], and a major difference between land plants and other complex organisms in that regard is the extreme plasticity of plant development. Because growth and stress responses are also tightly intertwined, it is possible that plants use a plethora m⁶A-YTHDF growth-stimulators in different organs, responsive to different stresses and developmental cues, to shape plant architecture in response to stimuli. The answer to the paradox may be in the details: plant YTHDFs responsive to different environmental cues may fine-tune growth programs through slightly different performance and target-binding capacity in partly overlapping expression domains, where they are known to compete for targets [23].

## Materials and methods

### Plant material and growth conditions

All the lines used in this study are in the *A. thaliana* Columbia-0 (Col-0) ecotype. The T-DNA insertion lines *rdr6-12* [72], *ect1-1* (SAIL_319_A08), *ect1-2* (GK-547H06), and *ect1-3* (SALK_059722) were obtained from Nottingham Arabidopsis Stock Centre. The mutants *ect2-3*, *ect3-2*, *ect2-3/ect3-2* (*Gde23*), *ect2-3/ect3-2/ect4-2* (*Gte234*) and *ect2-1/ect3-1/ect4-2* (*te234*) used for genetic crosses or as background to produce transgenic lines, and the primers for genotyping the corresponding alleles, have been previously described [16,17]. The standard growing conditions used are detailed in S1 Appendix.

### Phylogenetic analysis

Amino acid sequences of all YTHDF proteins found in 40 species distributed among the main taxa of Viridiplantae were downloaded from UniProt (Apweiler et al., 2004), TAIR (www. arabidopsis.org), Phytozome [107], GinkgoDB [108], FernBase [109] and PhycoCosm [110,111] (S1 Dataset). As outgroup, we selected three species from the eukaryotic taxa Glaucophyta (*Cyanophora paradoxa*, *Cpa*), Haptophyta (*Emiliania huxleyi*, *Ehu*) and Rhizaria (*Paulinella microscpora*, *Pmi*), as these are evolutionarily closer to Viridiplantae than e.g. Opisthokonta such as animals and fungi [99, 100]. All sequences were aligned with MUSCLE [112,113] using default parameters (S2 Dataset). The alignment was trimmed from both ends to discard regions with poor sequence conservation according to the quality and conservation scores provided by Jalview [114] (S3 Dataset and S23 Fig). The retained part of the alignment, comparable to the region used for phylogenetic analysis in a previous study [26], contains the entire canonical YTH domain [8] and, on both sides, short stretches with high conservation scores (S23 Fig). Evolutionary analyses were conducted using the maximum likelihood algorithm IQ-Tree 2 [115,116] from the CIPRES Science Gateway v. 3.3 (https://www.phylo.org/index.php/) to infer a phylogenetic tree. The model based on Q matrix [117] estimated for plants [118], 'Q.plant+R8', was determined as best-fit using the built-in tool ModelFinder [119] according to Bayesian Information Criterion (BIC) scores, and used for the analysis. Measures of branch supports by approximate Likelihood Ratio Test [aLRT] (SH-aLRT, %) [120] were also calculated. The resulting tree (S4 Dataset) was rooted on *Pmi* YTHDF, annotated adding SH-aLRT values, and color-coded using FigTree (https://github.com/rambaut/figtree/). An additional alignment performed with a subset of these proteins with the purpose to illustrate the conservation of the YTH domain in green algae and non-flowering plants was calculated and trimmed as described above (S5 Dataset). The graphical representations of this alignment (S3 and S4 Figs) were obtained using Jalview [114].

Estimates of evolutionary divergence between *A. thaliana* and *M. polymorpha* YTHDF proteins were obtained computing pairwise distances from the full length alignment of homologs in a subset of land plant species (bryophytes, basal angiosperms, magnoliids and dicots) (S6 Dataset). These taxa were chosen based on the observation that inclusion of divergent proteins from green algae, ferns, gymnosperms and monocots causes loss of information due to more frequent gaps, while proteins from species closely related to *M. polymorpha* (bryophytes) and *A. thaliana* (dicots and basal angiosperms) maximize the accuracy of the alignment. The analysis was done as described above, and we used the Equal Input model [121] in MEGA11 [122] to calculate the distances. All positions containing gaps and missing data were eliminated (complete deletion option), resulting a total of 171 positions in the final dataset. The complete matrix can be found in S7 Dataset.

## Cloning, plant transformation and line selection

We employed the scar-free USER cloning method [123] to produce *US7Yp:cYTHDF(X)-mCherry-OCSter* and *ECT1p:gECT1-FLAG-TFP-ECT1ter* constructs by gluing PCR-amplified DNA fragments into the pCAMBIA3300-U plasmid [124] that provides resistance to glufosinate in plants. Cloning of the trial *ECT2p:gECT1-mCherry-ECT2ter* and *ECT2p:gECT4-m-Cherry-ECT2ter* constructs was done with GreenGate [125]. The detailed cloning, transformation and line selection strategy for each construct, as well as the specific details of T1 selections for the complementation assay, are explained in S1 Appendix.

## IDR modeling and simulations

We carried out coarse-grained simulations of protein IDRs using the implicit-solvent CALVADOS 2 forcefield [78]. In CALVADOS, amino-acid residues are mapped to single beads corresponding to their amino-acid type. Neighboring residues are linked using harmonic bonds with an equilibrium distance of 0.38 nm and force constant of 8033 kJ mol$^{-1}$nm$^{-2}$. Nonbonded interactions are modeled with a hydropathy amino-acid "stickiness" model using $\lambda$-parameters from CALVADOS 2 [78,126]. Salt-screened electrostatic interactions are accounted for with a truncated ($r_c$ = 4 nm) and shifted Debye-Hückel potential; non-electrostatic interactions were truncated at 2 nm [126]. We used a Langevin integrator with friction coefficient $\gamma$ = 0.01 ps$^{-1}$ to propagate the simulation in 0.01 ps timesteps. All simulations were carried out at T = 293 K.

We performed slab simulations of IDRs of the following proteins: *Ath* ECT1-11, *Dm* YTHDF1, *Hs* YTHDF2 and FUS, *Mpo* DFE, *Cpu* DFE, *Ita* DFE1-3, *Cri* DFF1-3, DFAB and DFD1-2, *Gbi* DFF1-2, DFA1-2, DFB, DFC1-2 and DFD, and *Atr* DFA1, DFB, DFC1-2 and DFD, using a previously described procedure [126,127]. IDR sequences of all proteins are detailed in S1 Dataset. Briefly, we simulated 100 copies of each IDR (one type of IDR at a time) in a simulation slab of dimensions 17x17x300 nm for *Ath* ECT1/3/6-11, *Gbi* DFB and DFF1 and *Atr* DFC2, 15x15x150 nm for *Hs* FUS, or 25x25x300 nm for all other proteins, accounting for the different IDR sequence lengths. We simulated each system for 7500 ns and recorded configurations at 1-ns intervals. The first 2000 ns of each simulation were considered as equilibration and not used for analysis. We centered the protein condensates in the z-dimension of the simulation box during post-processing. The concentrations of the dense phase and dilute phase were calculated by fitting the concentration profiles to a hyperbolic function, as described by Tesei *et al.* [78]. Excess transfer free energies were calculated from the dilute and dense phase equilibrium concentrations via $\Delta G_{trans} = RT \ln (c_{dilute} / c_{dense})$ [128]. All simulation data and analysis scripts are available at the Zenodo repository (https://doi.org/10.5281/zenodo.8269719).

Charges (q) were added along the IDR sequences to obtain cumulative charge plots. Arg and Lys residues were assigned charges of q = +1; Glu and Asp residues were assigned charges of q = -1; other residue types, including His, were assigned charges of q = 0. The average 'stickiness' of a given IDR sequence was determined by calculating the mean hydrophobicity $\bar{\lambda}$ of the IDR amino acid composition, using the CALVADOS 2 hydrophobicity scale reported in [126].

## Additional methods

S1 Appendix describes additional standard methods including DNA and cDNA obtention, genotyping (see S13C–S13F Fig and S2 Table for additional details), fluorescence microscopy, qPCR (see S9 Dataset for the raw data), RNA and protein blotting, and the characterization of YTH domains.

## Supporting information

**S1 Fig. Fully annotated phylogenetic tree.** Same phylogenetic analysis of Fig 1, but with rectangular layout, and indicating the protein and species name of all YTHDFs, and aLRT values (%) for all nodes. The representative proteins labeled in Fig 1B are highlighted with larger typography on the right side of this figure as a reference.
(PDF)

**S2 Fig. YTH domain proteins in Archaeoplastida.** Number of genes encoding YTHDF and YTHDC proteins in several species of Archaeoplastida (see Figs 1 and S1 for detailed data on Embryophyta), and closely related taxa. NI, Not Identified. Phylogenetic relationships between groups according to [55,62,100,101] are indicated. Among chlorophytes, YTHDF proteins were found only in species of the *Micromonas* and *Bathycoccus* genera of the basal group Mamiellophyceae, although losses in other genera within this group have also occurred (e.g. *Ostreococcus* [68]). Regarding charophytes, the only sequenced species of their earliest-branching group, *Mesostigma viride*, has lost YTHDF proteins altogether, but YTHDF-encoding genes are found in four species of later-diverging groups: *Klebsormidium nitens*, *Chara braunii*, *Zygnema circumcarinatu*, and *Mesotaenium endlicherianum*.
(PDF)

**S3 Fig. Conservation of YTHDF proteins in green algae.** Amino acid sequence alignment of the YTH domains of DF proteins in the Viridiplantae species used for the phylogenetic analysis in Fig 1B, highlighting the family members found in green algae (see species abbreviations in S2 Fig). The frequent amino acid substitutions in structural elements of the green algal orthologs are revealed by the absence of the blue shading that indicates percentage identity (Jalview [114]). In some chlorophyte DFs, these substitutions affect the m⁶A-recognition aromatic cage (red arrows and red/yellow shading), because the two embryophyte-invariant tryptophan residues in the recognition loop (Fig 5A) exhibit changes to proline, histidine or phenylalanine. Chlorophyte DF proteins also contain insertions in their YTH domains, here marked with magenta lines and numbers above the sequences that correspond to collapsed gaps in S4 Fig. The insertions have variable lengths, reaching up to 77 amino acids, and can disrupt the m⁶A-recognition loop, e.g. the poly-Glycine stretch of *Micromonas commoda* DF (insertion 10, GGGGGGGGGGGGGGGGGGPR). In the charophyte DF orthologs, the YTH domains have intact aromatic cages and overall higher conservation than those of chlorophytes, but they also differ considerably from the land plant relatives. For example, the sequences of the β1 and α1 structural elements (Fig 5A) have degenerated in one of the three *Klebsormidium nitens* DF proteins (*Kni* DF3), and a 156-amino acid insertion protruding

from the plant-specific α2-extension is apparent in the only DF protein from *Mesotaenium endlicherianum* (*Men* DF), a species that belongs to the sister clade of all embryophytes (S2 Fig).
(PDF)

**S4 Fig. Conservation of YTHDF proteins in cryptogams and gymnosperms.** Amino acid sequence alignment of the YTH domains of DF proteins in plant taxa that diverged before the evolution of flowers. A few angiosperm YTHDFs are included in the analysis for comparison. Magenta-colored numbers at the top indicate the positions of amino acid insertions in one or few proteins. Because these insertions create gaps in the aligned sequences of the other homologs, they have been hidden to gain space and clarity, but the corresponding sequences can be found in S3 Fig and S5 Dataset. Red arrows under the alignment point to the conserved aromatic residues that contact $m^6A$, and a green arrow marks the Asp-to-Asn substitution, present in YTHDCs, that increases the affinity for $m^6A$ by 15-fold [61]. This substitution is also found in the plant DF-B clade [26], and here revealed in fern DF-F proteins as well. Blue-white coloring and the conservation and consensus tracks at the bottom are computed by Jalview [114] as in S3 Fig.
(PDF)

**S5 Fig. Expression of gDNA constructs of *ECT1*, *ECT2* and *ECT4* under the control of the *ECT2* promoter.** 9-day-old primary transformants expressing ECT1, ECT2 or ECT4 fused to mCherry in the *te234* background. All transgenes are expressed from gDNA under the control of the *ECT2* promoter. mCherry fluorescence on the right panels show the variability in transgene expression levels with this setup. Additional genotypes in the top panels are shown as a reference. Scale bars are 0.5 mm.
(PDF)

**S6 Fig. Expression of *cECT2-mCherry* driven by the ribosomal protein *US7Y* promoter can rescue late leaf emergence in *ect2-1/ect3-1/ect4-2* (*te234*) seedlings. (A)** Raw percentages of seedlings with first true leaves >0.5 mm at 10 days after germination among primary transformants of *te234* or *rdr6-12/te234* plants transformed with *US7Yp:cECT2-mCherry-OCSt* (cDNA) or *US7Yp:mCherry-OCSt* (control) compared to *ECT2p:gECT2-mCherry-ECT2t* (gDNA). **(B)** Expression pattern in 9-day-old seedlings of the indicated genotypes (T2 for *US7Yp*-driven constructs, and T5 for *ECT2p:gECT2-mCherry-ECT2t*). Fluorescence and protein abundance is typically higher in plants expressing free mCherry (*US7Yp:mCherry-OCSt*) than in fusions of mCherry with ECTs (see S8 and S15 Figs). Although the expression pattern of *US7Yp:cECT2-mCherry-OCSt* is identical to that of *ECT2p:gECT2-mCherry-ECT2t*, the fluorescence level and protein abundance observed is typically lower among cDNA lines in the *te234* background. Scale bars are 1 mm for images of aerial tissues (upper panels), and 0.1 mm for roots (lower panels).
(PDF)

**S7 Fig. Complementation rates (raw data). (A-D)** Absolute complementation rates for each independent transformation in *te234* (A, B, D) or *rdr6-12/te234* (C) plants given by the percentage of primary transformants (T1s) with first true leaves bigger than 0.5 mm at 10 days after germination for *US7Yp:cECT(X)-mCherry-OCSt* (A, C), *US7Yp:cECT(X)$_{IDR}$/cECT(Y)$_{YTH}$-mCherry-OCSt* chimeras (B, C), or *US7Yp:cYTHDF(X)-mCherry-OCSt* heterologous constructs (C, D). Numbers over the bars indicate the total number of transformants. Asterisks indicate absence of mCherry fluorescence (expression) among all retrieved transformants, or very faint fluorescence that underwent silencing in the next generation (T2). S6 Dataset contains detailed T1 counts and the calculation of complementation rates.
(PDF)

**S8 Fig. Expression of *US7Yp:cECT(X)-mCherry-OCSt* constructs in *te234* plants. (A)** Representative 10-day-old *te234/US7Yp:cECT(X)-mCherry-OCSt* T1 seedlings with their fluorescent signal and genetic background controls. Bars represent weighed averages of complementation over 2–5 transformations (Fig 2C). Scale bars are 1 mm. **(B)** Levels of protein expression from *US7Yp:cECT(X)-mCherry-OCSt* constructs in independent lines (L) of 9-day-old T2 seedlings assessed by α-mCherry western blot. All genotypes are in the *te234* background. Lines with single insertions and the highest complementation capacity (largest true leaves at 9 days after germination in T2) were selected for the analysis. ECT9 is not included because we could not observe fluorescence in the T2 generation for any of the few lines obtained, due to silencing. *te234 ECT2p:gECT2-mCherry-ECT2t* lines [16] are included in both membranes as a reference. Lanes marked with X in the lower membrane correspond to either lines irrelevant for this study, or left empty due to defects in the wells. Ponceau-staining is shown as loading control.
(PDF)

**S9 Fig. Transformation Efficiency.** Estimated transformation efficiency (average of 6 plates) of each construct for all the independent transformation batches of *te234* (TRF1-10) or *rdr6-12/te234* (TRF12) plants used in this study. TRF8 was screened twice (TRF8a and TRF8b). Red-to-blue colouring in the *te234* background reflects the variability in transformation efficiencies across constructs and batches, to highlight the consistently low recovery of transformants for *Arabidopsis thaliana* (*Ath*) ECT9 and *Homo sapiens* (*Hs*) YTHDF2.
(PDF)

**S10 Fig. Phenotypic characterization of selected transgenic lines in *rdr6-12/te234* and relevant genetic backgrounds. (A)** Developmental stages of plants of the indicated genotypes. DAG, days after germination. **(B)** Same as in A for primary transformants (T1s) of the indicating transgenes, all expressed from the *US7Y* promoter with mCherry fused at the C-terminus. Dashed outlines at 10 DAG are magnified below each panel to show mCherry fluorescence. Several different independent lines of the same age are shown for the genotypes exhibiting the strongest developmental defects. Plants in A and B were grown in parallel.
(PDF)

**S11 Fig. Expression of *US7Yp:cECT9-mCherry-OCSt* in the *rdr6-12* background has no obvious phenotypic effect.** Rosettes of 24-day-old primary transformants expressing the indicated transgenes in the *rdr6-12* background, grown in parallel with *rdr6-12* and *rdr6-12/te234* controls.
(PDF)

**S12 Fig. Expression levels of ECT paralogs in tissues of *Arabidopsis thaliana*.** Expression levels of ECT1-ECT11 across different tissues according to public mRNA-Seq data [73]. TPM, transcripts per million.
(PDF)

**S13 Fig. Isolation of *ECT1-TFP* transgenic lines and *ect1* T-DNA insertion alleles. (A)** Western blot using antibodies against GFP (that recognize TFP) in different *ECT1p:gECT1-TFP-ECT1t* independent lines. Ponceau (Ponc.) staining of the membrane is used as loading control. **(B)** Northern blot using the probe (P) specified in Fig 3G to detect *ECT1* mRNA. Although the probe recognizes specifically *ECT1* in the *ECT1p:gECT1-TFP-ECT1t* Line #21 (marked with an asterisk in A), the endogenous expression levels of *ECT1* in Col-0 wild type are below detection limit by northern blot. **(C)** Schematic representation of the *Arabidopsis thaliana ECT1* locus (At3g03950). Exons are represented as boxes and introns as lines.

Untranslated regions (UTRs) are coloured grey, the sequence encoding the YTH domain is purple, and the rest of the *ECT1*-coding sequence is black. The IDs and positions of the T-DNA insertions assigned to *ect1-1*, *ect1-2* and *ect1-3* alleles are marked, and so is the location of primers used for their genotyping. **(D-F)** 1% agarose gels showing EtBr-stained PCR fragments corresponding to the genotyping of *ect1-1* (D), *ect1-2* (E) and *ect1-3* (F) in plants germinated from seeds provided by the Nottingham Arabidopsis Stock Center (NASC) as indicated. The primer set for each PCR ('T-DNA' detects the insertion, and 'WT' detects the wild type allele) and the length of the resulting amplicons are indicated to the left of each gel. The sequence of all primers can be found in S2 Table. The progeny of plants homozygous for each T-DNA insertion, highlighted in shades of blue, was selected for crosses and further characterization.
(PDF)

**S14 Fig. Knockout of *ECT1* does not affect arabidopsis plant development. (A)** Abbreviations for combinations of double (d), triple (t), or quadruple (q) *ect* (e) mutants following the nomenclature proposed by Arribas-Hernández *et al.* [17]. The prefix 'G' identifies allele combinations having only T-DNA insertions belonging to the GABI-KAT collection [129]. **(B)** Morphological appearance of seedlings with or without *ECT1* in the different backgrounds indicated, at 9 or 17 days after germination (DAG). **(C)** Analyses of the percentage of trichomes with 3, 4, 5 or 6 spikes in plants with or without *ECT1* in the different backgrounds indicated. n, number of trichomes assessed for each genotype. NS, non-significant differences according to statistical analysis performed as in [16]. **(D)** Analysis of root growth rate and directionality upon mutation of *ECT1* in the *ect2-3* and *ect2-3/ect3-2* (*Gde23*) knockout backgrounds. Upper panels represent the overlayed silhouettes of actual roots as they grow on vertically disposed MS-agar plates. n, number of plants assessed for each genotype. VGI, vertical growth index; HGI, horizontal growth index. Root phenotypic analyses were conducted as in [17].
(PDF)

**S15 Fig. Expression of chimeric *ECT(X)$_{IDR}$-ECT(Y)$_{YTH}$* constructs in *te234* plants.** Representative 10-day-old *te234/US7Yp:cECT(X)$_{IDR}$-cECT(Y)$_{YTH}$-mCherry-OCSt* T1 seedlings with their fluorescent signal. Bars represent weighed averages of complementation. Scale bars are 1 mm.
(PDF)

**S16 Fig. Conservation of the YTH domain at the sequence level.** Sequence alignment of the YTH domains of *Arabidopsis thaliana* (*Ath*) ECT1-11, *Marchantia polymorpha* (*Mpo*) DFE, *Saccharomyces cerevisiae* (*Sc*) MRB1/Pho92, *Drosophila melanogaster* (*Dm*) YTHDF, and *Homo sapiens* (*Hs*) YTHDF2 and YTHDF1. The alignment is colored and annotated using ESPript 3 [130], according to the crystal structure of *Hs* YTHDF1 in complex with RNA GGm⁶ACU [61] as template for structural elements. The residues of *Hs* YTHDF1 that have contacts with RNA are marked above the sequences. Positions in *Ath* ECT1, ECT9 and ECT11 with non-conservative changes compared to the majority of ECTs are highlighted. Most of these substitutions are located on the protein surface according to 3D homology models of the ECT1/9/11 YTH domains (S17 Fig) except for the following residues: For **ECT1**, N260 and C261 (α1) are in close proximity to the aromatic cage. For **ECT9**, N432 (α1) forms hydrogen bonds with m⁶A. The D to N substitution at that position in all members of the DF-B clade (marked also in ECT5 and ECT10) [26], fern DF-Fs (S4 Fig) and YTHDC proteins [13, 61] increases the affinity for m⁶A by 15-fold [61] and is likely the product of convergent evolution. The remaining ECT9 substitutions are on the surface of the protein in areas not known to

interact with RNA except for S388 (*), also a serine in the DF-B member ECT5 (marked), despite the presence of glycine in this position in most YTHDC and YTHDF proteins of plants, animals and fungi. The backbone NH group of this glycine in *Hs* YTHDF1 (Gly444) and *Zygosaccharomyces rouxii* (*Zro*) MRB1 (Gly233) forms hydrogen bonds, through a water molecule, with the phosphate backbone of the RNA [11,61]. Remarkably, **ECT11** contains an arginine (R293) in the same position. Other ECT11 substitutions are on the surface and far from the RNA-binding groove. Of note, plant YTHDF proteins have a small insertion between α2 and β3 that extends α2 further out than in the metazoan orthologs. This insertion is slightly longer in the angiosperm DF-D and -C clades, which include *Ath* ECT11 (S17 Fig), as well as in all members of the fern/gymnosperm-exclusive DF-F clade (S4 Fig), and in many bryophyte and lycophyte DF-Es (S4 Fig).
(PDF)

**S17 Fig. Conservation of the YTH domain at the structural level.** Three left panels, AlphaFold-predicted [74,75] apo structures of YTH domains of *Arabidopsis thaliana* (*Ath*) ECT1/9/11 in comparison to that of *Ath* ECT2, and of *Homo sapiens* (*Hs*) YTHDF1. Red/blue coloring in the left-most panel indicates the electrostatic potential of the surface calculated using the APBS PyMOL plugin [131] with the parameters described in Methods. The two middle panels show the similarity of the secondary structure of all proteins and highlight the few non-conservative amino acid substitutions identified in *Ath* ECT1/9/11 (S16 Fig). Right panel, hydrophobic cage of the same proteins modelled on the crystal structure of the YTH domain of *Hs* YTHDF1 in complex with RNA GGm⁶ACU [61] (Protein Data Bank entry 4RCJ). The key residues for m⁶A binding are marked, including the aspartic to asparagine substitution in the DF-B clade member *Ath* ECT9 (N342).
(PDF)

**S18 Fig. Predicted structure and short linear motifs (SLiMs) of arabidopsis ECTs compared to human YTHDF2. Left columns**: Structure of *Arabidopsis thaliana* (*Ath*) ECT proteins predicted by AlphaFold [74,75] colored according to a confidence score (pLDDT) given per-residue between 0 and 100. Regions below 50 pLDDT may be unstructured in isolation [75, 132]. The vast majority of N-terminal regions in ECT proteins score below 50 pLDDT (dark orange). A few short stretches with low-confidence prediction of secondary structure are marked with black arrows for ECT1/9/11. Middle column: Disorder prediction by MobiDB [77] depicted by red (disorder) or light blue (structure) coloring of the AlphaFold-predicted structure. The N-terminal regions of all ECT proteins are predicted to be largely disordered (red). **Right columns**, sequences and linear representation of the proteins indicating MobiDB [77] disorder predictions ('Disorder' track) with the same colors as in the middle column. The known structure of the YTH domain of *Homo sapiens* (*Hs*) YTHDF2 is marked in dark blue, and the YTH domains of *Ath* ECTs, with no experimentally resolved structures to date, have been manually added also in dark blue. Purple bars superimposed on the amino acid sequences of the proteins and purple boxes positioned along the linear representations on the 'Linear interacting peptide' (LIP) tracks highlight residues predicted to interact with another molecule that preserve structural linearity in the bound state. In addition to LIPs, they are generally called short linear motifs (SLiMs), molecular recognition features (MoRFs) or protean segments (ProS). The UniProt ID of all proteins is given below their names.
(PDF)

**S19 Fig. Analysis of the IDRs of the YTHDF proteins assayed in this study (extended data). (A)** Time-averaged density profiles of the slab simulations of the ECT proteins shown in Fig 5B. **(B)** Relationship between excess transfer free energy from dilute to dense phase

($\Delta G_{trans}$ = RT ln [$c_{dilute}$ / $c_{dense}$]) and average stickiness ($\lambda$) of the IDR residues. **(C-D)** Comparative amino acid composition (C) and representation of charges (D) along the IDR sequences of the different YTHDF proteins assayed in this study. In D, arrows highlight the overall trends in charge change along the IDRs. Notice that the length in amino acids (aa) is not at scale to simplify the representation.
(PDF)

**S20 Fig. Heterologous expression of bryophyte and metazoan YTHDF proteins in arabidopsis (extended data). (A)** T1 plants expressing the long or short splice forms of *Marchantia polymorpha* (*Mpo*) DFE 19 days after germination (DAG). Additional genotypes are included as a reference. **(B)** Morphological appearance and fluorescence pattern of T2 plants expressing *Mpo* DFE$_L$ (L, long isoform) or *Homo sapiens* (*Hs*) YTHDF2 in the *te234* background 9 or 17 DAG. Additional genotypes are included as a reference. For *Hs* YTHDF2, several plants are shown to reflect the variability in phenotypic defects, which include enhanced aberrance in the shape or number of cotyledons and first true leaves, and slower overall growth. C, cotyledons; L, true leaves. White scale bars are 1 cm, and black are 1 mm.
(PDF)

**S21 Fig. Analysis of the IDRs of YTHDF proteins from representative species of the main taxa of land plants. (A)** Representation of charges (top panels) and slab simulations (bottom panels) of the IDR sequences of YTHDF proteins from the main taxa of land plants. The proteins are sorted according to land plant evolution (vertical axis) and plant DF clades (horizontal axis). *Isoetes taiwanensis* (*Ita*) DF-CD (Fig 1B) is excluded from the analysis because its N-terminus is very short (S1 Dataset) and does not behave like an IDR. **(B)** Relationship between excess transfer free energy from dilute to dense phase ($\Delta G_{trans}$ = RT ln [$c_{dilute}$ / $c_{dense}$]) and average stickiness ($\lambda$) of the IDR residues of the proteins in A. The proteins are color-coded according to plant DF clades. *Homo sapiens* (*Hs*) YTHDF2 is included as a reference. Same as for the *Arabidopsis thaliana* (*Ath*) ECT set in S19B Fig, the content of sticky residues defined by the CALVADOS model [78] mainly governs the different phase separation propensity among the IDRs. A clade-dependent separation of the proteins according to these properties is apparent.
(PDF)

**S22 Fig. Analysis of domain composition of YTHDF proteins from the charophyte *Klebsormidium nitens* (*Kni*). (A-C)** Prediction of domains and other features in the protein sequences of *Kni* DF1-3 according to InterPro (former Pfam) (https://www.ebi.ac.uk/interpro/) [133]. Horizontal axes indicate protein length in amino acids (aa). InterPro entries (unique protein homologous superfamilies, families, domains, repeats or important sites based on one or more signatures) are indicated on the right side. Disorder prediction is engineered by MobiDB (https://mobidb.bio.unipd.it/) [77], and transmembrane domains are predicted by Phobius (https://phobius.sbc.su.se/) [134]. For *Kni* DF2 in B, the graphical output of Phobius transmembrane topology predictor is shown.
(PDF)

**S23 Fig. Trimming of the alignment of Viridiplantae YTHDF proteins according to conservation for phylogenetic analyses. (A-B)** Overview of the full-length amino acid sequence alignment of Viridiplantae YTHDF proteins (S2 Dataset) (A), and magnification of the trimmed region used for phylogenetic analyses (S3 Dataset) (B). The conservation score provided by Jalview [114] is shown below both panels, and the threshold used for trimming is marked with a horizontal dotted line in A. Blue-white colouring reflects percentage identity according to Jalview [114], with more intense shades of blue for the most highly conserved residues. Grey indicates gaps. In A, the trimmed region is coloured green, with a darker shade

highlighting the position of the canonical YTH domain [8]. In B, the YTH domain is framed with a dashed dark green outline, and the aromatic residues that conform the m⁶A-recognition cage are coloured red and marked with red arrows.
(PDF)

**S1 Appendix. Extended Materials and Methods.**
(DOCX)

**S1 Table. Isoforms contained in cYTHDF constructs, and templates used for PCR.**
(XLSX)

**S2 Table. Constructs and DNA Oligonucleotides.** All sequences are 5' to 3'.
(XLSX)

**S1 Dataset. Protein Sequences.** Amino acid sequences of proteins used in this study with their identifiers and taxonomic classification. Abbreviations of species names match with those used in Figs 1 and S1. For YTHDF proteins, the canonical YTH domain is colored in blue, and long internal insertions within the YTH domain are highlighted in a paler shade of blue. Additional folded domains are marked in other colors. The sequence of the IDR used for biophysical simulations of some of the proteins is the entire sequence upstream to the YTH domain. For the human fused in sarcoma (FUS) RNA-binding protein used as control for the analyses of IDR properties, the IDR sequence is highlighted in **bold**.
(XLSX)

**S2 Dataset. Full length alignment of Viridiplantae YTHDF proteins.** Full-length amino acid sequence alignment of the YTHDF proteins used for the phylogenetic analyses in Figs 1 and S1. The alignment was obtained using MUSCLE (Multiple Sequence Comparison by Log-Expectation [112, 113]), and is given in clustal format.
(ALN)

**S3 Dataset. Trimmed alignment of Viridiplantae YTHDF proteins.** Amino acid sequence alignment of the YTHDF proteins used for the phylogenetic analyses in Figs 1 and S1 after trimming the poorly conserved regions (IDRs) at both ends, according to the score given by Jalview [114] (S23 Fig), from the alignment provided in S2 Dataset.
(ALN)

**S4 Dataset. Phylogenetic tree of Viridiplantae YTHDF proteins.** Phylogenetic tree (Maximum likelihood phylogenetic analysis) in Nexus/Newick format obtained from the trimmed alignment provided in S3 Dataset. Values of nodes are SH-aLRT (%) support.
(NEX)

**S5 Dataset. Full length alignment of YTHDF proteins in Viridiplantae focused on early-diverging clades.** Full-length amino acid sequence alignment of selected YTHDF proteins, with focus on early-diverging taxa within Viridiplantae, used for S3 and S4 Figs. The names of the proteins are preceded by a 'tag' related to the DF clade and organism taxon to facilitate a meaningful sorting of the alignment for illustration purposes. The alignment was obtained using MUSCLE (Multiple Sequence Comparison by Log-Expectation [112, 113]), and is given in clustal format.
(ALN)

**S6 Dataset. Raw T1 complementation counts.** Numerical data and calculations supporting the histograms in Figs 2C–2D, 4B, 6B–6C, and S7.
(XLSX)

**S7 Dataset. Full length alignment of YTHDF proteins for the calculation of evolutionary divergence of *Ath* ECTs.** Full-length amino acid sequence alignment of 101 YTHDF proteins from selected taxa (bryophytes, basal angiosperms, magnoliids and dicots) used for pairwise comparisons to calculate evolutionary distances between *Mpo* DFE and *Ath* ECTs (S8 Dataset, Fig 6A). All the sequences with their identifiers and the taxonomic classification of the organisms can be found in S1 Dataset. The alignment was computed with MUSCLE (Multiple Sequence Comparison by Log-Expectation [112,113]), and is given in clustal format. (ALN)

**S8 Dataset. Evolutionary divergence of *Ath* ECTs.** Matrix with estimates of evolutionary divergence (number of amino acid substitutions per site) between 101 land plant proteins from selected taxa (bryophytes, basal angiosperms, magnoliids and dicots) according to the alignment in S7 Dataset. The analysis was conducted using the Equal Input model [121] in MEGA11 [122,135]. All positions containing gaps and missing data were eliminated (complete deletion option). There were a total of 171 positions in the final dataset. A complete matrix ('Matrix output') as well as a selection of the comparisons regarding *Ath* ECTs vs. *Mpo* DFE, is extracted from the matrix in a separate sheet ('Selection'). Color-coding follows the rest of the figures in the article. The graphic representation of the values is shown in Fig 6A. (XLSX)

**S9 Dataset. Raw data and calculations supporting qPCR data in Fig 3H.** (XLSX)

**S1 Movie. ECT2 simulation.** Coarse-grained molecular dynamics simulation of 100 ECT2 IDRs using the CALVADOS 2 forcefield (7.5 microseconds simulation time) [126]. The proteins are colored differently to distinguish individual chains. (MP4)

**S2 Movie. ECT1 simulation.** Coarse-grained molecular dynamics simulation of 100 ECT1 IDRs using the CALVADOS 2 forcefield (7.5 microseconds simulation time) [126]. The proteins are colored differently to distinguish individual chains. (MP4)

**S3 Movie. ECT11 simulation.** Coarse-grained molecular dynamics simulation of 100 ECT11 IDRs using the CALVADOS 2 forcefield (7.5 microseconds simulation time) [126]. The proteins are colored differently to distinguish individual chains. (MP4)

## Acknowledgments

We thank Lena Bjørn Johansson, Sergio D'Anna, Jakub Najbar and Tobias Lahti for technical assistance, Theo Bølsterli and his team for plant care, and the Danish National Life Science Supercomputing Center Computerome and the core facility for biocomputing at the Department of Biology (University of Copenhagen) for the use of computational resources. We acknowledge the constructive criticism of Rupert Fray as a reviewer on previous publications, as his suggestion to use the *ect2/ect3/ect4* triple mutant phenotype to test ECT molecular function by systematic ectopic expression in the ECT2 expression domain inspired us to embark on the present project.

## Author Contributions

**Conceptualization:** Sören von Bülow, Kresten Lindorff-Larsen, Peter Brodersen, Laura Arribas-Hernández.

**Data curation:** Daniel Flores-Téllez, Mathias Due Tankmar, Sören von Bülow, Laura Arribas-Hernández.

**Formal analysis:** Daniel Flores-Téllez, Mathias Due Tankmar, Sören von Bülow, Laura Arribas-Hernández.

**Funding acquisition:** Sören von Bülow, Kresten Lindorff-Larsen, Peter Brodersen.

**Investigation:** Daniel Flores-Téllez, Mathias Due Tankmar, Sören von Bülow, Laura Arribas-Hernández.

**Methodology:** Daniel Flores-Téllez, Mathias Due Tankmar, Sören von Bülow, Junyu Chen, Laura Arribas-Hernández.

**Project administration:** Peter Brodersen.

**Resources:** Kresten Lindorff-Larsen, Laura Arribas-Hernández.

**Software:** Sören von Bülow.

**Supervision:** Kresten Lindorff-Larsen, Peter Brodersen, Laura Arribas-Hernández.

**Validation:** Sören von Bülow, Kresten Lindorff-Larsen.

**Visualization:** Daniel Flores-Téllez, Sören von Bülow, Laura Arribas-Hernández.

**Writing – original draft:** Peter Brodersen, Laura Arribas-Hernández.

**Writing – review & editing:** Daniel Flores-Téllez, Mathias Due Tankmar, Sören von Bülow, Kresten Lindorff-Larsen, Peter Brodersen, Laura Arribas-Hernández.

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
