## [Decision Letter · Decision Letter 0]

25 Jun 2023

Dear Dr Arribas-Hernández,

Thank you very much for submitting your Research Article entitled 'Conservation and diversification of the molecular functions of YTHDF proteins' to PLOS Genetics.

The manuscript was fully evaluated at the editorial level and by three independent peer reviewers. Although all the three reviewers appreciated the attention to an important problem, they, in particular Reviewer 2 and 3, also raised some substantial concerns about the current manuscript. Based on the reviews, we will not be able to accept this version of the manuscript, but we would be willing to review a much-revised version. We cannot, of course, promise publication at that time.

If you decide to revise the manuscript for further consideration at PLOS Genetics, please aim to resubmit within the next 60 days, unless it will take extra time to address the concerns of the reviewers, in which case we would appreciate an expected resubmission date by email to plosgenetics@plos.org.

We are sorry that we cannot be more positive about your manuscript at this stage. Please do not hesitate to contact us if you have any concerns or questions.

Yours sincerely,

Li-Jia Qu

Section Editor

PLOS Genetics

Reviewer's Responses to Questions

**Comments to the Authors:**

Reviewer #1: Adenosine methylation in mRNA is known to play important roles in plant development, and the YTH (plant ECT) family of proteins are known to be the main “readers” that bind to RNAs containing m6A. However, much confusion and lack of information exists around how the binding of such proteins brings about normal growth. The eukaryote YTH family can be further divided into nuclear YTHDC-type and cytoplasmic YTHDF-type readers. In both plants and animals, it is the YTHDF-type that have been evidenced as being most critical for bringing about the majority of the m6A-dependent post transcriptional outcomes. In humans there are three YTHDFs, and roles in promoting mRNA translation, degradation or stabilization have variously been claimed, though there is a growing consensus that all promote destabilisation. Higher land plants have an expanded family of YTHDFs, and determining whether these all carry out the same function or have different (perhaps even opposing) functions is a pressing question in the field, which this paper seeks to address.

Overall, the manuscript is very well written, the experiments are clear and easy to follow and there are several important findings that will be of considerable interest to the field. Most notably;

1) The fact that the single YTHDF found in liverworts can rescue the ECT2/3/4 phenotype is a critical finding as it indicates a conserved and ancient function for basal plant YTHDF function.

2) The finding that ECT1 does not rescue ECT2/3/4 is unexpected and appears to be primarily due to altered functionality of the disordered N terminus. This might have been due to more complex regulation of ECT1, such as requirement for post translational modification. However, this is argued against by the lack of additional growth defect in the quadruple mutant.

3) The complementation by at least some members of each of the 4 ECT sub groups suggests that the primordial function has been retained. This suggests that the reason for expansion of this family in plants may reflect differences in expression/regulation of the different ECTs rather than evolution of different functions upon mRNA binding. However, the lack of complementation by ECT9, ECT1 and ECT11 is curious. The elegant domain swap experiments indicate that this is likely due to neofunctionalisation of the disordered domains.

4) The enhancement of the triple mutant phenotype following expression of ECT9 is particularly striking, especially as this seems to be dependent upon the presence of the existing te234 mutations. This may suggest an antagonistic function in this case and this is likely to become an active area of research.

The human YTHDF2 also enhanced the te234 phenotype - likely due to competition for m6A binding, but remains an interesting and potentially useful observation.

5) The combination of complementation experiments, domain swap-complementation and (in the case of ECT9) inhibition, taken together with the known requirement of ECT2’s m6A binding for functionality, implies that ALL of the Arabidopsis ECTs are capable of minding m6A transcripts – an important and highly significant finding.

6) The structural analysis of the disordered domains suggests that charge distribution and ability to form droplets may be playing an important role in determining ECT function as a te234 complimenter.

7) The improved phylogenetic tree is a useful addition to the field and now clearly shows the division of higher plant ECTs into the four sub groups.

Overall, this is a thorough and rigorous piece of work that begins to answer one of the key questions in the plant mRNA methylation field and opens new avenues for research. Open questions remain, particularly around the mRNA targets of ECT9 (and ECT1 and ECT11), and developmental effects such as trichome branching (strongly influenced by m6A level and the te234 mutation), but these will likely be addressed in follow up work.

Reviewer #2: This manuscript reveals that “conservation and diversification of the molecular functions of YTHDF proteins” is mainly introduced that the m6A/YTHDF axis controls cellular proliferation and organ growth in plants, the same molecular functions that allow ECT2/3/4 to regulate stem cell proliferation. Lineage-specific neo-functionalization of ECT1, ECT9 and ECT11 happened after late duplication events. The expression pattern of ECT5/6/7/8/10 may be used in specific biological contexts response to stimuli, suggesting different molecular activities of YTHDF proteins. This manuscript needs improvements before it could be considered for publication.

First, the data in this manuscript are roughly descriptive. Detailed molecular mechanisms are still lacking. A possible way to improve is to show how the different YTHDF readers bind to m6A to determine cytoplasmatic properties of target transcripts.

Second, the presentation is suggested to be improved. Here are some specific suggestions:

1) Figure1 “Phylogenetic analysis of YTHDF proteins in land plants” should mark more YTHDF proteins information about Arabidopsis, the paper mainly uses Arabidopsis to carry out research.

2) The paper includes some unreported mutants, such as “ect1-1 (SAIL_319_A08), ect1-2 (GK-547H06), ect1-3 (SALK_059722), ect2-3, ect3-2, ect2-3/ect3-2 (Gde23), ect2-3/ect3-2/ect4-2 (Gte234) and ect2-1/ect3-1/ect4-2”, the author can show some identification of mutants in supplemental data.

3) In Fig. 3H, there is no difference significance analysis for the columns, the bar value of column is a little high, it is recommended to repeat this experiment.

4) The font size of line 150 and line 151 is different, please use the same font size.

5) The reference in line 685 of the article is incorrectly cited.

6) The line 854 “Tesei G, Lindorff-Larsen K, 2022.” should be changed to “Tesei G, Lindorff-Larsen K (2022)”.

7) In the middle of title of citing references should be a lowercase letter, such as the lines 644, 647,660…

8) Many formatting errors in this manuscript. It is suggested that this manuscript should carefully proofread.

Reviewer #3: This study conducted phylogenetic analyses of YTHDF proteins, and performed a series of functional complementation assay, to investigate its molecular function in plants. The authors showed that YTHDF homolog was present in the early-diverging land plants and remained conserved over its diversification. The authors mainly focused on YTHDF protein paralogs in A. thaliana, and less efforts were paid on YTHDF protein homologs in other representative species in the main lineages of green plants. The complement assays of YTHDF protein paralogs in A. thaliana is insufficient for the interpretation of function conservation or diversification, and it would be better to investigate the gene expression and phenotypes in corresponding mutants. There are some concerns that needed to be clarified.

1.The results section should be carefully checked and rephrased:

(1) combine some results and reduce the use of subheadings;

(2) reduce the description of previous results, for example, Line 116-125;

(3) check the result and move the descriptions of method to the method section, for example, Line 162-170.

2.The manuscript aimed to investigate the molecular function of plant YTHDF proteins. However, the taxa samplings of plant genomic data appear to be deficiency, only containing few land plants and ignoring the early-branching green plants (such as chlorophytes and charophytes).

3. For the phylogenetic analyses, why did the authors trimmed from their N-termini to discard intrinsically disordered regions with poor sequence conservation. In general, the protein sequences of homologs should be aligned, and then the multiple sequence alignments should be trimmed with proper threshold values.

4. For the complement assays, the authors mainly focused on YTHDF protein paralogs in A. thaliana, and few evidence were shown to elucidate the protein evolution in plants. It would be great interests to determine the molecular functions of YTHDF protein homologs in several representative species in the main lineages of green plants, including chlorophytes, charophytes, bryophytes, ferns, gymnosperms.

5. This section, “The ability to complement ect2/3/4 does not simply follow affiliation with phylogenetic clades”, is of interests, but lack extensive discussions about this pattern.

**Have all data underlying the figures and results presented in the manuscript been provided?**

Reviewer #1: Yes

Reviewer #2: None

Reviewer #3: Yes

PLOS authors have the option to publish the peer review history of their article (what does this mean?). If published, this will include your full peer review and any attached files.

Reviewer #1: **Yes: **Rupert Fray

Reviewer #2: No

Reviewer #3: No

---

## [Decision Letter · Decision Letter 1]

17 Sep 2023

Dear Dr Arribas-Hernández,

We are pleased to inform you that your manuscript entitled "Insights into the conservation and diversification of the molecular functions of YTHDF proteins" has been editorially accepted for publication in PLOS Genetics. Congratulations!

Yours sincerely,

Li-Jia Qu

Section Editor

PLOS Genetics

Li-Jia Qu

Section Editor

PLOS Genetics

Comments from the reviewers (if applicable):

Reviewer's Responses to Questions

**Comments to the Authors:**

Reviewer #1: This is an important research paper that clarifies a long standing question in the plant epitranscriptomics field. It will be of interest to, and will influence the work of, many m6A researchers, and thus it is important that it should be published and made available to the community as a priority.

Reviewer #2: no comments

**Have all data underlying the figures and results presented in the manuscript been provided?**

Reviewer #1: None

Reviewer #2: Yes

PLOS authors have the option to publish the peer review history of their article (what does this mean?). If published, this will include your full peer review and any attached files.

Reviewer #1: **Yes: **Rupert Fray

Reviewer #2: No

**Data Deposition**

http://datadryad.org/submit?journalID=pgenetics&manu=PGENETICS-D-23-00522R1

**Press Queries**

---

## [Editor Report · Acceptance letter]

5 Oct 2023

PGENETICS-D-23-00522R1 

Insights into the conservation and diversification of the molecular functions of YTHDF proteins 

Dear Dr Arribas-Hernández, 

We are pleased to inform you that your manuscript entitled "Insights into the conservation and diversification of the molecular functions of YTHDF proteins" has been formally accepted for publication in PLOS Genetics! Your manuscript is now with our production department and you will be notified of the publication date in due course.

With kind regards,

Zsofi Zombor

PLOS Genetics

On behalf of:
